# Understanding the Variance Collapse of SVGD in High Dimensions

**Jimmy Ba[1,2], Murat A. Erdogdu[1,2], Marzyeh Ghassemi[1,2], Shengyang Sun[1,2],**
**Taiji Suzuki[3,4], Denny Wu[1,2], Tianzong Zhang[5]**

[1]University of Toronto, [2]Vector Institute, [3]University of Tokyo, [4]RIKEN AIP, [5]Tsinghua University
{jba, erdogdu, marzyeh, ssy, dennywu}@cs.toronto.edu,
taiji@mist.i.u-tokyo.ac.jp, ztz16@mails.tsinghua.edu.cn

## Abstract

Stein variational gradient descent (SVGD) is a deterministic inference algorithm that evolves a set of particles to fit a target distribution. Despite its computational efficiency, SVGD often underestimates the variance of the target distribution in high dimensions. In this work we attempt to explain the variance collapse in SVGD. On the qualitative side, we compare the SVGD update with gradient descent on the maximum mean discrepancy (MMD) objective; we find that the variance collapse phenomenon relates to the bias from deterministic updates present in the "driving force" of SVGD, and empirically verify that removal of such bias leads to more accurate variance estimation. On the quantitative side, we demonstrate that the variance collapse of SVGD can be accurately predicted in the proportional asymptotic limit, i.e., when the number of particles $n$ and dimensions $d$ diverge at the same rate. In particular, for learning high-dimensional isotropic Gaussians, we derive the exact equilibrium variance for both SVGD and MMD-descent, under certain empirically verified near-orthogonality condition on the converged particles, and confirm that SVGD suffers from the "curse of dimensionality".

## 1 Introduction

A typical challenge in Bayesian learning is to efficiently and accurately learn a complex posterior distribution. Markov Chain Monte Carlo (Robert and Casella, 2013) provides asymptotically accurate samples, but simulating the chain till convergence can be time-consuming. On the other hand, variational inference (Wainwright et al., 2008) approximates the intractable target distribution with parametric families by minimizing the KL divergence, which can be more computationally efficient. Among a plethora of variational approximations, particle-based inference algorithms are particularly compelling as they bypass the inherent distributional assumptions on the variational family.

As a noticeable example, Stein Variational Gradient Descent (SVGD) (Liu and Wang, 2016) is a deterministic particle-based inference algorithm that iteratively transports the particles by the functional gradient of KL divergence in the reproducing kernel Hilbert space (RKHS). The functional gradient takes the form of a kernelized Stein's operator and only requires access to the unnormalized target density. Despite the empirical successes of SVGD (Liu, 2017; Haarnoja et al., 2017; Kim et al., 2018), very few convergence guarantees have been established except in the mean-field limit, i.e., the number of particles $n \to \infty$ under fixed dimensionality $d$, for which it has been shown that the distribution of particles converges to the true invariant solution (Liu, 2017; Lu et al., 2019). Moreover, it has been observed that as the problem dimensionality $d$ becomes larger, the variance estimated by SVGD can be much smaller than variance of the target distribution (Zhuo et al., 2017). This observation, which we refer to as the *variance collapse* phenomenon (see Figure 1), is highly undesirable for practitioners due to two reasons: $(i)$ underestimating the variance leads to failure in explaining the uncertainty of model predictions, which is a key benefit of being Bayesian; $(ii)$ modern Bayesian inference problems are usually high-dimensional; for instance, training Bayesian neural networks (BNNs) (MacKay, 1992) requires inferring the posterior distribution of network weights, which could be more than millions of dimensions in real-world problems (Krizhevsky et al., 2012).

**Our Contribution.** We study the algorithmic bias of SVGD that leads to variance collapse in high dimensions. We focus on the commonly-used Euclidean distance kernel (e.g., Gaussian RBF kernel), and provide: $(i)$ qualitative understanding on the pitfall of SVGD by comparing it to another interacting particle algorithm termed *MMD-descent*; $(ii)$ quantitative understanding on the variance underestimation in learning simple target distributions. Our findings can be summarized as follows.

- In Section 3 we connect SVGD with MMD-descent, a kernel-based particle inference algorithm that performs maximum mean discrepancy (MMD) minimization, and empirically show that despite their similar updates, MMD-descent does not collapse the variance in high dimensions.
- In Section 4 we identify the log derivative driving force as the problematic term in SVGD, and design experiments to illustrate that the combination of this driving force and the *deterministic bias*, i.e. the absence of particle resampling, prevents accurate variance estimation in high dimensions.
- In Section 5 we argue that the *proportional asymptotic limit*: $n, d \to \infty$, $d/n \to \gamma \in (0, \infty)$, is more relevant to understanding the variance collapse phenomenon. As an example, we derive the exact dimension-averaged variance of SVGD and MMD-descent in learning Gaussian distribution in the proportional limit, under certain concentration assumption which we empirically verify. Our analysis confirms that variance estimated by SVGD scales *inversely* with the dimensionality.

## 2 BACKGROUND

### 2.1 INTEGRAL PROBABILITY METRIC

To measure the "closeness" between distributions on $\mathcal{X} \subseteq \mathbb{R}^d$, one may consider the maximum discrepancy between the target $p$ and sample distribution $q$ over test functions $\mathcal{F}$: $D_{\mathcal{F}}(p, q) := \sup_{h \in \mathcal{F}} \mathbb{E}_q[h(\boldsymbol{x})] - \mathbb{E}_p[h(\boldsymbol{y})]$, known as the *integral probability metric* (IPM) (Müller, 1997).

If $\mathcal{F}$ is a unit ball in the reproducing kernel Hilbert space (RKHS) $\mathcal{H}$, the resulting $D_{\mathcal{F}}$ is termed the *maximum mean discrepancy* (MMD) (Gretton et al., 2012), and its squared value can be evaluated as:

$$\text{MMD}^2(p, q) = \mathbb{E}_{\boldsymbol{x},\boldsymbol{x}'}[k(\boldsymbol{x}, \boldsymbol{x}')] + \mathbb{E}_{\boldsymbol{y},\boldsymbol{y}'}k(\boldsymbol{y}, \boldsymbol{y}') - 2\mathbb{E}_{\boldsymbol{x},\boldsymbol{y}}[k(\boldsymbol{x}, \boldsymbol{y})] \tag{1}$$

where $\boldsymbol{x}, \boldsymbol{x}' \sim p$, $\boldsymbol{y}, \boldsymbol{y}' \sim q$, and the kernel $k : \mathbb{R}^d \times \mathbb{R}^d \to \mathbb{R}$ satisfies $\mathbb{E}\sqrt{k(\boldsymbol{x}, \boldsymbol{x})} < \infty$.

**Stein's Discrepancy.** When integration under the target $p$ is intractable, *Stein's method* (Stein et al., 1972) can be used to construct zero-mean test functions w.r.t. $p$. For differentiable $h$ in the Stein Class of $p$, i.e., $\int_x \nabla_{\boldsymbol{x}}(h(\boldsymbol{x})p(\boldsymbol{x}))d\boldsymbol{x} = 0$, the Stein's discrepancy (Gorham and Mackey, 2015) is given as

$$D_{\mathcal{F}}^{\text{Stein}}(p, q) = \sup_{h \in \mathcal{F}} \mathbb{E}_q[h(\boldsymbol{x})^{\top}\nabla_{\boldsymbol{x}} \log p(\boldsymbol{x}) + \nabla_{\boldsymbol{x}}^{\top}h(\boldsymbol{x})].$$

Note that the Stein's discrepancy only involves the score of $p$ and thus the normalization constant is not required. When $h$ is restricted in the product RKHS $\mathcal{H}^d$ with inner product $\langle f, g \rangle_{\mathcal{H}^d} = \sum_{i=1}^d \langle f_i, g_i \rangle_{\mathcal{H}}$, the corresponding maximum discrepancy, known as *kernel Stein discrepancy* (KSD), can be estimated efficiently from samples (Liu et al., 2016; Chwialkowski et al., 2016).

### 2.2 (DETERMINISTIC) PARTICLE INFERENCE ALGORITHMS

We now consider the approximation of $p$ using particles constructed as follows: starting from the initial particles $X = \{\boldsymbol{x}_i\}_{i=1}^n$, we iteratively optimize their positions using the update: $\boldsymbol{x}_i = \boldsymbol{x}_i + \eta\Delta(\boldsymbol{x}_i)$, where $\eta$ is the step size, and $\Delta(\cdot) : \mathbb{R}^d \to \mathbb{R}^d$ is the update direction. Motivated by the SVGD algorithm, we focus on the setting where the update $\Delta$ is *deterministic*, in which case the particles may converge to a deterministic fixed point. Note that there exist other particle-based algorithms that do not fit into this description, such as sequential Monte Carlo (Doucet et al., 2001), and stochastic variants of the particle gradient descent method (Chen et al., 2018a; Gallego and Insua, 2018), which our current analysis does not cover (see Appendix A.4 for discussion on these alternative algorithms).

**Stein Variational Gradient Descent.** SVGD constructs the update direction $\Delta$ as the optimal perturbation in the RKHS that decreases Kullback-Leibler divergence. In particular, constrain $\Delta$ in RKHS unit ball, and take $q = \frac{1}{n}\sum_{i=1}^n \delta_{\boldsymbol{x}_i}$, the update for each particle $\boldsymbol{x}$ is given as:

$$\Delta^{\text{SVGD}}(\boldsymbol{x}) = \mathbb{E}_{\boldsymbol{x}' \sim q}[\underbrace{k(\boldsymbol{x}', \boldsymbol{x})\nabla_{\boldsymbol{x}'} \log p(\boldsymbol{x}')}_{\text{driving force}} + \underbrace{\nabla_{\boldsymbol{x}'}k(\boldsymbol{x}', \boldsymbol{x})}_{\text{repulsive force}}] := \frac{1}{n}\sum_{i=1}^n [S_1(\boldsymbol{x}_i, \boldsymbol{x}) + S_2(\boldsymbol{x}_i, \boldsymbol{x})]. \tag{2}$$

Intuitively, the *log derivative* term in the update rule $S_1(\boldsymbol{x}_i, \boldsymbol{x}) := k(\boldsymbol{x}_i, \boldsymbol{x})\nabla_{\boldsymbol{x}_i} \log p(\boldsymbol{x}_i)$ corresponds to a *driving force* that guides particles towards high likelihood regions, whereas the *kernel derivative* term $S_2(\boldsymbol{x}_i, \boldsymbol{x}) := \nabla_{\boldsymbol{x}_i} k(\boldsymbol{x}_i, \boldsymbol{x})$ provides a *repulsive force* to prevent the particles from collapsing into the mode. Typically-used kernels in SVGD include the Gaussian RBF kernel $k(\boldsymbol{x}, \boldsymbol{x}') = \exp(-\|\boldsymbol{x} - \boldsymbol{x}'\|_2^2 / 2\sigma^2)$ (Liu and Wang, 2016; Zhuo et al., 2017) and the inverse multi-quadratic (IMQ) kernel $k(\boldsymbol{x}, \boldsymbol{x}') = 1/\sqrt{1 + \|\boldsymbol{x} - \boldsymbol{x}'\|_2^2/(2\sigma^2)}$ (Gorham and Mackey, 2017).

## 3 CONNECTING SVGD WITH MMD MINIMIZATION

In this section, we introduce another particle inference algorithm termed *MMD-descent*, whose update rule closely relates to that of SVGD. Despite the similarity, we empirically observe MMD-descent accurately estimates the target variance independent of the dimensionality. By analyzing the similarity of the update rules and contrast in the algorithmic performances, we identify factors that lead to the variance collapse of SVGD. In this work we focus on the following class of kernel functions.

**Definition 1.** *A **Euclidean Distance Kernel** can be written as:* $k(\boldsymbol{x}, \boldsymbol{y}) = f\left(\|\boldsymbol{x} - \boldsymbol{y}\|_2^2/\sigma^2\right)$.

For example, the Gaussian RBF kernel and IMQ kernel both satisfy this definition. Note that $\sigma$ is the tunable bandwidth that usually scales with the distance between particles.

We now construct a kernel-based particle inference algorithm termed MMD-descent, which draws inspiration from the kernel herding algorithm (Welling, 2009) and particle gradient descent (Chizat and Bach, 2018). Instead of greedily reducing the MMD by adding one particle a time, we consider an update rule similar to SVGD that transport all particles together to approximate the target distribution, by minimizing the MMD between the particles and the target via gradient descent:

$$\Delta^{\text{MMD}}(\boldsymbol{x}) = \mathbb{E}_{\boldsymbol{y} \sim p}[\underbrace{\nabla_{\boldsymbol{x}} k(\boldsymbol{x}, \boldsymbol{y})}_{\text{driving force}}] + \mathbb{E}_{\boldsymbol{x}' \sim q}[\underbrace{-\nabla_{\boldsymbol{x}} k(\boldsymbol{x}, \boldsymbol{x}')}_{\text{repulsive force}}].$$

This update can be seen as the finite-particle discretization of the MMD gradient flow (Arbel et al., 2019). For Euclidean distance kernels we have $-\nabla_{\boldsymbol{x}} k(\boldsymbol{x}, \boldsymbol{x}') = \nabla_{\boldsymbol{x}'} k(\boldsymbol{x}, \boldsymbol{x}') = S_2(\boldsymbol{x}', \boldsymbol{x})$; hence, MMD-descent and SVGD share the same repulsive force. Furthermore, when $k$ is in the Stein class of $p$, the driving force of MMD-descent can be connected SVGD via a simple integration by parts:

$$\mathbb{E}_{\boldsymbol{y} \sim p}[\nabla_{\boldsymbol{x}} k(\boldsymbol{x}, \boldsymbol{y})] = -\mathbb{E}_{\boldsymbol{y} \sim p}[\nabla_{\boldsymbol{y}} k(\boldsymbol{x}, \boldsymbol{y})] = \mathbb{E}_{\boldsymbol{y} \sim p}[k(\boldsymbol{x}, \boldsymbol{y})\nabla_{\boldsymbol{y}} \log p(\boldsymbol{y})] = \mathbb{E}_{\boldsymbol{y} \sim p}[S_1(\boldsymbol{y}, \boldsymbol{x})]. \quad (3)$$

Therefore, the MMD-descent update for a set of particles $\{\boldsymbol{x}_i\}_{i=1}^n$ can be equivalently written as:

$$\Delta^{\text{MMD}}(\boldsymbol{x}) = \mathbb{E}_{\boldsymbol{y} \sim p}[S_1(\boldsymbol{y}, \boldsymbol{x})] + \frac{1}{n}\sum_{i=1}^n S_2(\boldsymbol{x}_i, \boldsymbol{x}). \quad (4)$$

We remark that MMD-descent is not a practical algorithm due to the required integration under $p$, which is typically intractable. Instead, the purpose of introducing MMD-descent is to compare the update with SVGD and understand the cause of the variance collapse phenomenon.

**SVGD vs. MMD-descent.** By comparing SVGD (2) and MMD-descent (4), we observe that:

- SVGD and MMD-descent have identical *repulsive force*.

- In MMD-descent, the *driving force* is integrated under the target distribution $p$, whereas in SVGD the expectation is under the current particle distribution $q$.

At the infinite particle limit, $p = q$ is a fixed point for both update rules. In fact, under certain conditions, uniqueness of this fixed point has been established for both updates in this mean-field limit (Lu et al., 2019; Arbel et al., 2019). However, in the practical setting where the particle size is not significantly greater than the dimensionality, it is not clear if the two algorithms: ($i$) provide a reasonable "approximation" to the target distribution; ($ii$) converge to similar fixed points. Given the analogous updates, one natural question to ask is: do SVGD and MMD-descent reliably estimate the target variance in high dimensions, and do they converge to similar solutions?

The empirical answer is in the negative: SVGD and MMD-descent converge to completely different solutions: SVGD is known to underestimate the marginal variance in high dimensions, even for simple Gaussian targets (Zhuo et al., 2017). In contrast, MMD-descent does not exhibit the same issue empirically. Figure 1 illustrates this discrepancy in a Bayesian neural network (BNN) experiment.

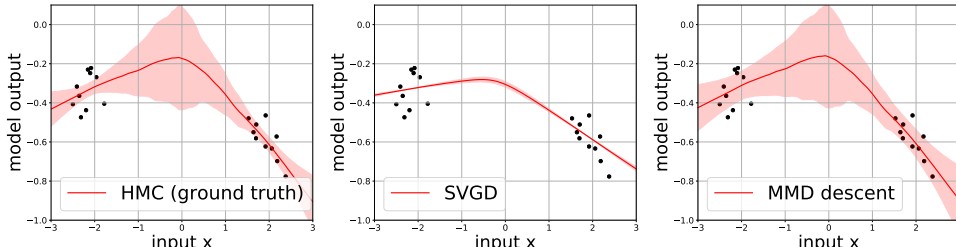

Figure 1: Comparison of SVGD and MMD-descent in training a two-hidden-layer BNN on synthetic 1D dataset. The target distribution is approximated via HMC, and 100 particles is used for SVGD and MMD-descent. SVGD (middle) significantly underestimates the target variance, but MMD-descent (right) generates diverse samples.

We consider a two-hidden-layer BNN with 100 hidden units each, so that inference in the weight space is a high-dimensional problem. We obtain (approximate) target samples via Hamiltonian Monte Carlo (HMC) (Neal et al., 2011), and then initialize 100 particles from the target to be optimized by SVGD or MMD-descent with the Gaussian RBF kernel (for detailed setup see Appendix D). Observe that although we used the same number of particles for SVGD and MMD-descent, SVGD severely underestimates the variance, whereas MMD-descent generates diverse predictions similar to HMC.

## 4 UNDERSTANDING THE PITFALL OF SVGD

We now qualitatively characterize the algorithmic bias of SVGD. Motivated by the observation that SVGD and MMD-descent only differs in the driving force which involves $S_1$, we first compare $S_1$ and $S_2$ in SVGD and argue that the former term is more likely problematic in high dimensions due to larger fluctuation. We then design controlled experiments by modifying the updates of SVGD and MMD-descent to show that *when the log derivative $S_1$ is coupled with the deterministic bias (i.e., absence of particle resampling), the algorithm does not reliably estimate the target variance.*

### 4.1 HIGH VARIANCE FROM INTEGRATION BY PARTS

Recall that the update rule of SVGD involves a driving force (*log derivative*) term $\frac{1}{n}\sum_{i=1}^{n} S_1(\boldsymbol{x}_i, \boldsymbol{x})$ and a repulsive force (*kernel derivative*) term $\frac{1}{n}\sum_{i=1}^{n} S_2(\boldsymbol{x}_i, \boldsymbol{x})$. From integration by parts we have the following equality: $\mathbb{E}_{\boldsymbol{y}\sim p}[S_1(\boldsymbol{y}, \boldsymbol{x})] = -\mathbb{E}_{\boldsymbol{y}\sim p}[S_2(\boldsymbol{y}, \boldsymbol{x})]$. Thus the target distribution $p$ is a fixed point of SVGD when the number of particles tends to infinity. However, in the finite sample setting, the fluctuations in $S_1$ and $S_2$ might have impact on the converged solution.

In many settings, we expect the log derivative driving force $S_1$ to have higher variance. For instance, in the case of unit Gaussian target $p$ and Gaussian RBF kernel $k(\boldsymbol{x}, \boldsymbol{x}') = \exp(-\|\boldsymbol{x}-\boldsymbol{x}'\|_2^2/2\sigma^2)$, we know that $S_1(\boldsymbol{y}, \boldsymbol{x}) = -\boldsymbol{y}k(\boldsymbol{y}, \boldsymbol{x})$ and $S_2(\boldsymbol{y}, \boldsymbol{x}) = \sigma^{-2}(\boldsymbol{x}-\boldsymbol{y})k(\boldsymbol{y}, \boldsymbol{x})$. Roughly speaking, when the norm of $\boldsymbol{x}, \boldsymbol{y}$ and the bandwidth $\sigma$ scale with $\sqrt{d}$, $\|S_1\|_2^2$ increases with the dimensionality, whereas $\|S_2\|_2^2$ remains bounded. The large magnitude of $S_1$ results in large fluctuation in the driving force term in SVGD. In Figure 2(a), we visualize the distributions of $S_1$ and $S_2$ when the particles $\boldsymbol{x}$ are drawn i.i.d. from Gaussian distributions, and indeed observe the higher variance of $S_1$. The following proposition quantifies this discrepancy in an idealized setting.

**Proposition 2.** *Define the mean squared error as:* $\mathrm{MSE}_p[f(\boldsymbol{y})] = \mathbb{E}_{p(\boldsymbol{y})}\|f(\boldsymbol{y}) - \mathbb{E}_p[f(\boldsymbol{y})]\|_2^2$. *Then for* $\boldsymbol{y} \sim \mathcal{N}(\boldsymbol{a}, I_d)$ *where* $\|\boldsymbol{a}\|_2^2 = O(d)$, *Gaussian RBF kernel* $k(\boldsymbol{x}, \boldsymbol{y}) = \exp(-\|\boldsymbol{x}-\boldsymbol{y}\|_2^2/2\sigma^2)$ *with bandwidth* $\sigma = \Theta(\sqrt{d})$, *and* $\boldsymbol{x} \in \mathbb{S}^{d-1}(\sqrt{d})$, *we have*

$$\mathrm{MSE}_p[S_2(\boldsymbol{y}, \boldsymbol{x})] = \Theta(d^{-1}); \ \ \mathrm{MSE}_p[S_1(\boldsymbol{y}, \boldsymbol{x})] = \Theta(d).$$

*More generally, for* $\boldsymbol{x} \in \mathbb{S}^{d-1}(\sqrt{d})$, *strongly log-concave* $p$, *and Euclidean distance kernel with Lipschitz* $f$ *and lower-bounded by a scaled Gaussian kernel,* $\mathrm{MSE}_p[S_1(\boldsymbol{y}, \boldsymbol{x})] = \Omega(d) \cdot \mathrm{MSE}_p[S_2(\boldsymbol{y}, \boldsymbol{x})]$.

The proposition indicates that when particles are i.i.d. sampled from some target $p$, then $S_1$ would have much larger fluctuation than $S_2$[1]. Intuitively speaking, the higher variance in $S_1$ suggests that more samples are required to estimate the term accurately. Therefore, when the dimensionality is large compare to the particle size, poor estimation of $S_1$ may relate to the variance collapse. The following subsection empirically demonstrate that this is indeed the case; in particular, we show that in the presence of *deterministic bias*, the high-variance $S_1$ leads to "biased" particles in SVGD.

---

[1]Note that particles optimized by SVGD cannot be considered as i.i.d. samples from some distribution, and thus Proposition 2 does not rigorously apply to particles obtained by the actual algorithm.

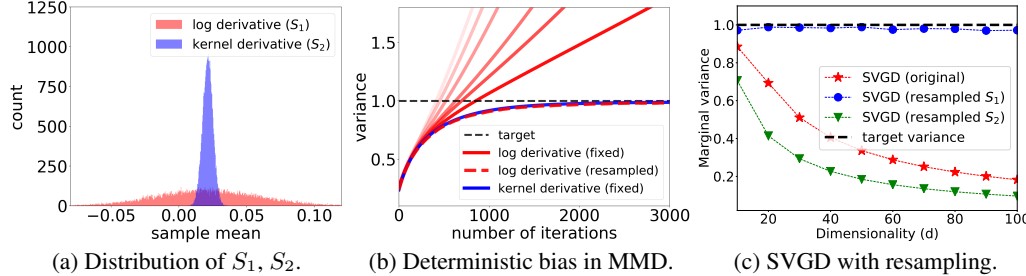

(a) Distribution of $S_1, S_2$.     (b) Deterministic bias in MMD.     (c) SVGD with resampling.

Figure 2: (a) Distribution of sample means of $S_1$ and $S_2$ for Gaussian target and Gaussian RBF kernel. (b) Particle variance of MMD-descent initialized from $\mathcal{N}(0, 0.2I_d)$. Darker color indicates larger particle size. Resampling or $\Delta_2^{\mathrm{MMD}}$ (fixed) results in correct variance, but $\Delta_1^{\mathrm{MMD}}$ (fixed) diverges. (c) SVGD (n=20) with resampled $S_1$ accurately estimates the variance, but resampled $S_2$ still leads to variance collapse.

## 4.2 Bias from Deterministic Update

In this subsection we isolate a cause of variance collapse that relates to the deterministic nature of the update rule. Observe that in the derivation of SVGD, particles $\{x_i\}_{i=1}^n$ are assumed to be sampled from some underlying distribution $q$. But due to the deterministic update, $q$ is entirely represented by the *same set of particles*, and redrawing i.i.d. samples is not feasible during optimization. We now demonstrate that this failure to resample, which we refer to as the *deterministic bias*, when combined with the log derivative driving force $S_1$, may cause the algorithm to converge to the wrong target.

**Deterministic Bias in MMD-descent.** We start with an illustration of deterministic bias in MMD-descent. Given samples $\{y_i\}_{i=1}^m \sim p$ drawn from the target, we have two distinct MMD-gradient updates (connected via integration by parts (3)) that differs in the driving force.

$$\Delta_1^{\mathrm{MMD}}(x) = \frac{1}{m}\sum_{i=1}^m S_1(y_i, x) + \frac{1}{n}\sum_{i=1}^n S_2(x_i, x). \quad \Delta_2^{\mathrm{MMD}}(x) = -\frac{1}{m}\sum_{i=1}^m S_2(y_i, x) + \frac{1}{n}\sum_{i=1}^n S_2(x_i, x).$$

We refer to $\Delta_1^{\mathrm{MMD}}$ and $\Delta_2^{\mathrm{MMD}}$ as the *log derivative* update and the *kernel derivative* update, respectively. Section 4.1 suggests that $\Delta_1^{\mathrm{MMD}}$ tend to have larger fluctuation than $\Delta_2^{\mathrm{MMD}}$. For both update rules, we consider the two variations: ($i$) *fixed* particles, where we draw $\{y_i\}_{i=1}^m$ in the beginning and use this same set of target samples throughout optimization; ($ii$) *resampled* particles, where we redraw new i.i.d. samples $\{y_i\}_{i=1}^m$ from $p$ at each iteration. In this construction, the fixed-particle update ($i$) emulates the *deterministic bias* in $S_1$, due to the absence of resampling from $p$.

We simulate the two variants of MMD-descent in Figure 2(b), in which the target is a 50-dimensional unit Gaussian, $n = 50$ and we vary $m$ from 50 to 1000. First observe that the *log derivative* update with fixed particles (solid red), which combines the high variance $S_1$ and the deterministic bias, results in diverged particles. In contrast, the *log derivative* update with resampled particles (dashed red), in which $S_1$ is present but not the deterministic bias, converges to the desired variance. Moreover, if the driving force is estimated by $S_2$, then the variance is accurate even in the presence of deterministic bias, as shown in the *kernel derivative* update with fixed particles (solid blue). This indicates that the combination of deterministic bias and the log derivative driving force $S_1$ leads to biased convergence[2].

**Particle Resampling in SVGD.** The previous experiment on MMD-descent suggests that the deterministic bias arises from the algorithm not being able to "redraw" i.i.d. samples. We further validate this observation in SVGD by constructing a variant that achieves particle resampling. The modified update exhibits a double-loop structure (see Algorithm 1): at iteration $i$ of the outer loop, we obtain a new set of particles indexed as $q_i$ after $(i-1)$ inner loop steps. In the inner loop, we first draw $n$ i.i.d. particles $\hat{q}_i$ from the initial distribution $q_0$, and then update $\hat{q}_i$ via $(i-1)$ SVGD steps to obtain $q_i$; at each

---

**Algorithm 1** SVGD with Particle Resampling

**Input:** Initial density $q_0$. Number of Steps $S$.
**for** $s = 1$ **to** $S$ **do**
   Sample $\{x_i^s\}_{i=0}^n \sim q_0(x)$.
   **for** $t = 1$ **to** $s - 1$ **do**
      **either** Compute $S_1$ from resampled particles:
         $x_i^s \leftarrow x_i^s + \frac{\eta}{n}\sum_{j=1}^n S_1(x_j^t, x_i^s) + S_2(x_j^s, x_i^s)$.
      **or** Compute $S_2$ from resampled particles:
         $x_i^s \leftarrow x_i^s + \frac{\eta}{n}\sum_{j=1}^n S_1(x_j^s, x_i^s) + S_2(x_j^t, x_i^s)$.
   **end for**
**end for**
Output $\{x_i^S\}_{i=1}^n$.

---

step $j$ of the inner loop, *either* the driving force $S_1$ or the repulsive force $S_2$ is computed between $\hat{q}_i$ and $q_j$, as opposed to between particles within $\hat{q}_i$ as in the original SVGD.

---

[2]Due to the different driving force, the log derivative update of MMD-descent with fixed particles results in divergence instead of variance collapse (see Appendix A.3 for more discussion).

We remark that this resampled update resembles transport-based particle algorithms (e.g. Nitanda and Suzuki (2017)), where old particles define a transport map that the new particles (initialized from $q_0$) follow. By construction, at each outer loop iteration, a new set of independent particles are generated and updated; importantly, one of $S_1$ and $S_2$ is *not* evaluated on the same set of particles themselves, and hence deterministic bias is not present. However, the complexity of such resampled updates scales *quadratically* with the iterations, which renders the algorithm computationally prohibitive[3].

Figure 2(c) compares the modified SVGD updates (with Gaussian RBF kernel) in learning a Gaussian target. Observe that when the driving force $S_1$ is computed on resampled particles (blue), SVGD accurately estimates the target variance even with small particle size ($n = 20$) at each step. In contrast, if only the repulsive force $S_2$ is evaluated on particles resampled from $q_0$, then the variance is still underestimated (green), due to deterministic bias in $S_1$. This confirms our argument that *combination* of log derivative $S_1$ and deterministic bias leads to the algorithmic bias of SVGD.

## 5 SVGD IN THE PROPORTIONAL LIMIT

While the previous discussion provides qualitative understanding of the algorithmic bias of SVGD, no quantitative characterization is provided. In this section, we make use of the *deterministic* nature of the SVGD update, and directly analyze the particle fixed point; this allows us to precisely compute the variance of SVGD particles and confirm the variance collapse in simple settings.

### 5.1 SCALING LIMIT AND BASIC ASSUMPTIONS

Previous works have considered the infinite-particle limit of SVGD under fixed dimensionality, which is known as the *mean-field limit* (Lu et al., 2019; Duncan et al., 2019). In this regime, the particle dynamics of SVGD is described by a nonlinear PDE: $\partial_t \rho = \nabla \cdot (\rho(K * (\nabla \rho + \nabla V \rho)))$, which converges weakly to the unique target measure as $t \to \infty$ under certain non-degeneracy assumptions. However, since the invariant solution is the desired target distribution, the mean-field limit does not capture the variance collapse phenomenon. Moreover, similar guarantees can also be derived for MMD-descent under similar assumptions (Arbel et al., 2019). Therefore, the mean-field limit does not explain the observed discrepancy between SVGD and MMD-descent (Figure 1 in Section 3).

Instead, we propose to analyze the algorithm in the *proportional asymptotic limit*:

- **(A1) Proportional Limit.** $n, d \to \infty$ and $d/n \to \gamma$; we also assume $\gamma > 1$.

Note that larger $\gamma$ is analogous to higher dimensionality or smaller particle size; we focus on the $\gamma > 1$ regime, where the number of particles is less than the problem dimensionality — this is a common feature of modern Bayesian inference problems. In this asymptotic limit, prior works studied spectral properties of the kernel matrix (e.g., El Karoui et al. (2010)), which the SVGD update crucially depends upon. However, these random matrix results typically require the particles to be i.i.d., which is not satisfied by SVGD due to the *interacting* update. To overcome this technical difficulty, we make an additional assumption on the fixed point:

- **(A2) Near-orthogonality.** The optimized particles $\{\boldsymbol{x}_i\}_{i=1}^n$ satisfy $|\boldsymbol{x}_i^\top \boldsymbol{x}_i - dv| < v\epsilon_d$, $|\boldsymbol{x}_i^\top \boldsymbol{x}_j| < v\epsilon_d$ for all $i \neq j$, some $v > 0$, and $d^{-1/2}\epsilon_d \to 0$ as $d \to \infty$ with probability 1.

The above assumption ensures that the particles tend to equilibrate at a fixed point where they "repel" one another as much as possible (due to the repulsive force), as shown in Figure 3 for the case of Gaussian target. While this is a condition only on the fixed point of the SVGD algorithm, not on the particles produced by the SVGD iterations (unless initialized at the fixed point), we note that it is still stronger than making an assumption on the sampling problem. We refer to Appendix A.2 for additional empirical verification and leave the rigorous justification as future work.

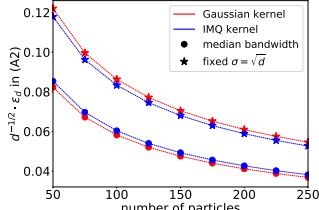

Figure 3: Empirical verification of (A2) for SVGD in learning isotropic Gaussian ($\gamma = 3$).

Under assumptions (A1) and (A2), we can approximate the Gram matrix via Taylor expansion of the kernel matrix, which allows us to simplify the fixed point equation. We restrict ourselves to the following class of kernels:

---

[3]Resampling can also be achieved via kernel density estimation (KDE) at each iteration, as in Dai et al. (2016). However, KDE is also known to be less effective in high dimensions.

- **(A3) Kernel Function.** $k(\boldsymbol{x}, \boldsymbol{y}) = f\left(\frac{\|\boldsymbol{x}-\boldsymbol{y}\|_2^2}{2\sigma^2}\right)$, where $f : \mathbb{R}_{\geq 0} \to \mathbb{R}_+$ is bounded, monotonically decreasing, and differentiable[4] on $\mathbb{R}_{\geq 0}$.

**Remark.** *(A3) covers common choices of Euclidean distance kernels with decaying tail, such as the Gaussian RBF kernel, the IMQ kernel and the log-inverse kernel. Our analysis also allows for different choices of bandwidth $\sigma$, which we specify in the sequel.*

## 5.2 LEARNING HIGH-DIMENSIONAL GAUSSIANS

We now consider a simple example in which SVGD does not reliably estimate the variance – learning an isotropic Gaussian target (empirically observed in Zhuo et al. (2017)). Note that our goal is to demonstrate a *negative result*: we do not claim that SVGD underestimates the variance in all settings; instead, we show that variance collapse is present even in learning the simplest target distribution.

- **(A4) Gaussian Target.** $p(\boldsymbol{x}) \propto \exp\left(-\frac{1}{2}\boldsymbol{x}^\top \boldsymbol{x}\right)$.

**Remark.** *We assume (A4) for simple and concise presentation, and we believe that the result can be extended to more general sub-Gaussian targets. In addition, since Gaussian prior is widely used in Bayesian inference, it is natural to expect that target potentials in many real-world problems are "Gaussian-like" (i.e., exhibit quadratic growth) outside of some radius (Cheng et al., 2018a). Indeed, our empirical observations on BNN align with our theoretical predictions for Gaussian target.*

We compute the *dimension-averaged marginal variance*: $v = \frac{1}{d}\sum_{j=1}^d \mathrm{Var}_j(\{\boldsymbol{x}_i\}_{i=1}^n)$, where $\mathrm{Var}_j(\cdot)$ is the particle variance at the $j$-th coordinate. Since $v$ is a scalar quantity, accurate estimation in the proportional limit (A1) is not impossible; in fact, in Appendix A.4 we show that running Langevin Monte Carlo (LMC) for $\mathcal{O}(d)$ iterations using *one particle* would suffice. In the following subsections we derive the equilibrium variance of SVGD under two different choices of kernel bandwidth.

**Kernels with Adaptive Bandwidth.** We first consider the case where the bandwidth $\sigma$ is adaptively tuned based on the optimized particles. In particular, we analyze the median heuristic (Scholkopf and Smola, 2001): $\sigma = \sqrt{\mathrm{Med}\{\|\boldsymbol{x}_i - \boldsymbol{x}_j\|_2^2\}/2}$, which is the most common choice made in practice. Under the previous assumptions, we have the following result for SVGD with median bandwidth.

**Proposition 3.** *Given (A1-4) and the choice of bandwidth $\sigma = \sqrt{\mathrm{Med}\{\|\boldsymbol{x}_i - \boldsymbol{x}_j\|_2^2\}/2}$, particles driven by SVGD (Equation (2)) equilibrate at the dimension-averaged variance*

$$v^{SVGD} = \gamma^{-1} \cdot f'(1)\left[f(1) - f(0)\right]^{-1}.$$

**Remark.** *The proposition precisely captures our observation that SVGD underestimates the dimension-averaged variance inversely proportional to the problem dimensionality ($\gamma = d/n$). In contrast, previous analysis (Zhang et al., 2020) predicts that SVGD would collapse to zero variance as $t \to \infty$, which does not align with the empirical observations.*

For Gaussian kernel, we also provide a precise contrast between SVGD and MMD-descent.

**Corollary 4.** *Given (A1-4), for the Gaussian RBF kernel $f(x) = exp(-x)$ with the median bandwidth, SVGD converges to the following dimension-averaged variance: $v^{SVGD} = (e-1)^{-1} \cdot \gamma^{-1} < 1$, whereas MMD-descent (Equation (4)) leads to $v^{MMD} = 1$.*

In other words, as the problem dimensionality $\gamma$ increases, more particles are required for SVGD to reliably estimate the dimension-averaged variance (specifically, the number of particles should grow linearly with $d$). In contrast, the variance estimated by MMD-descent remains accurate and is independent to $\gamma$; this aligns with the empirical observations in Section 3. Remarkably, Figure 4(a) demonstrates that Proposition 3 is accurate even for reasonably small $n, d$: observe that once $\gamma > 1$ (i.e., $d > n$), the prediction (black) becomes well-aligned with the empirical value.

**Kernels with Fixed Bandwidth.** One possibility remains that the culprit of variance collapse is the median bandwidth. Specifically in kernel Stein discrepancy (KSD), it has been argued that the IMQ kernel can outperform the Gaussian RBF kernel (Gorham and Mackey, 2017), and the IMQ kernel is often employed without an adaptive bandwidth. Due to the connection between KSD and SVGD, one may speculate that the IMQ kernel with fixed bandwidth can alleviate the variance collapse problem.

---

[4]For the median bandwidth, we only require local differentiability around 1, similar to El Karoui et al. (2010).

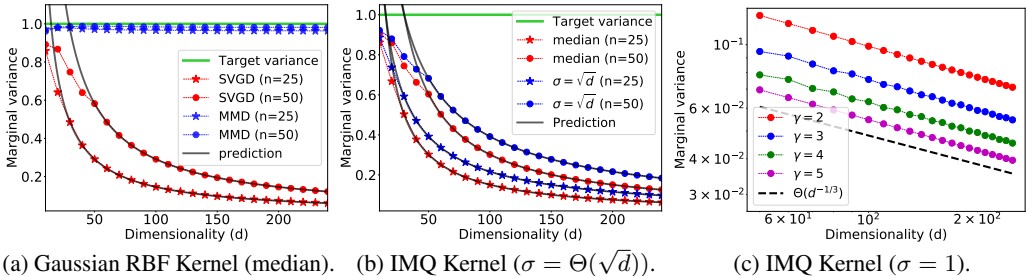

(a) Gaussian RBF Kernel (median).   (b) IMQ Kernel ($\sigma = \Theta(\sqrt{d})$).   (c) IMQ Kernel ($\sigma = 1$).

Figure 4: Stationary variance of SVGD and MMD-descent; predictions (black) are given by Proposition 3 and 5. (a) Gaussian kernel with median heuristic: SVGD underestimates the variance, but MMD-descent (blue) does not. (b) IMQ-SVGD underestimates the variance under both the median heuristic (red) and fixed $\sigma = \sqrt{d}$ (blue). (c) When $\sigma = 1$, IMQ-SVGD asymptotically collapses the variance to 0 at a rate of $d^{-1/3}$ (black).

We show that this is unfortunately not the case. We first provide a general characterization:

**Proposition 5.** *Given (A1-4) and fixed bandwidth[5] $\sigma = \sqrt{d}$, the SVGD variance satisfies*
$$f'(v^{SVGD}) = \gamma \cdot \left[f(v^{SVGD}) - f(0)\right].$$
*In addition, if $f'$ is also monotone on $\mathbb{R}_{\geq 0}$, then $v^{SVGD}$ decreases as $\gamma > 1$ increases.*

Note that the monotonicity assumption is again satisfied by many Euclidean distance kernels of interest. While the equation may not provide an explicit expression of the stationary variance, we can verify that the variance *decreases* as the problem becomes more high-dimensional (i.e., larger $\gamma$). Our next proposition specifically handles the IMQ kernel considered in Gorham and Mackey (2017).

**Corollary 6.** *Given (A1-4), for the IMQ kernel $f(x) = (1 + x)^{-1/2}$ with fixed bandwidth, we have the following stationary variance of SVGD under two different scalings:*

- *When $\sigma = \sqrt{d}$, $v^{SVGD} < 1$ and is decreasing as $\gamma > 1$ increases.*
- *When $\sigma = 1$, $v^{SVGD} \to 0$ as $n, d \to \infty$ at a rate of $d^{-1/3}$.*

This corollary suggests that the IMQ kernel with fixed bandwidth is not a remedy to the variance collapse problem. We remark that the second setting ($\sigma = 1$) is considered in Gorham and Mackey (2017) for KSD. In both cases (and including the median bandwidth covered by Proposition 3), SVGD-IMQ underestimates the target variance as the dimensionality increases (large $\gamma$). The agreement between the theoretical predictions and the empirical simulations (using finite particles) is illustrated in Figure 4(b)(c). Finally, in Appendix A we include a more general (but less precise) characterization of variance collapse beyond Gaussian target, as well as additional empirical evidence.

### 5.3   A MODIFICATION OF SVGD

We have thus far shown that in the simple setting of learning high-dimensional Gaussian, SVGD underestimates the dimension-averaged variance unless the number of particles is larger than the dimensionality. Now we further validate our theoretical findings by introducing a heuristic modification of SVGD that corrects for this variance collapse in the overparameterized regime[6].

The starting observation is that the variance collapse indicates that the deterministic bias causes the *driving force* term to dominate. Because during each update every particle $\boldsymbol{x}_i$ is most "correlated" with itself, one should expect $S_1(\boldsymbol{x}_i, \boldsymbol{x}_i)$ to contribute significantly to this bias. We thus consider a modification of SVGD which simply shrinks (damps) the term $S_1(\boldsymbol{x}_i, \boldsymbol{x}_i)$ by $\lambda = \min\{1, (f(1) - \gamma^{-1}f'(1))/f(0)\}$, where $\lambda$ is chosen such that when $d > n$, the equilibrium variance matches the target variance in the setup of Proposition 3. We refer to this update as *damped SVGD*:

$$\Delta^{\text{Damp}}(\boldsymbol{x}_i) = \sum_{j \neq i} [S_1(\boldsymbol{x}_j, \boldsymbol{x}_i) + S_2(\boldsymbol{x}_j, \boldsymbol{x}_i)] + \lambda S_1(\boldsymbol{x}_i, \boldsymbol{x}_i). \tag{5}$$

**Remark.** *At high level, (5) resembles the annealed SVGD algorithm (D'Angelo and Fortuin, 2021), in that they both "weaken" the driving force $S_1$. However, annealed SVGD uses a heuristic learning rate schedule (target-independent) to modify the strength of $S_1$, whereas we derive our $\lambda$ to ensure that the dimension-averaged variance is correct under (A1-4) in the $d > n$ regime.*

---

[5]Note that the fixed bandwidth can be arbitrary, as constants can be absorbed into the function $f$.

[6]The modified update in Section 4 also alleviate the variance collapse, but is computationally intractable.

**Simulating Toy Densities.** We demonstrate the usefulness of this bias-correction term in learning simple target distributions. We stack 250 independent 2-dimensional distributions to create a high-dimensional target distribution, and visualize the first two dimensions in Figure 5. We optimize 50 particles using SVGD or SVGD (damped) with the Gaussian RBF kernel and median bandwidth.

As shown in Figure 5 (top), SVGD collapses its particles to the mode, whereas damped SVGD generates more dispersed particles with variance exactly matching that of the Gaussian target. It is also possible that the heuristic modification leads to more accurate samples for other "Gaussian-like" target distributions[7], as shown in the second row of Figure 5. We provide additional experimental results in Appendix A.3.

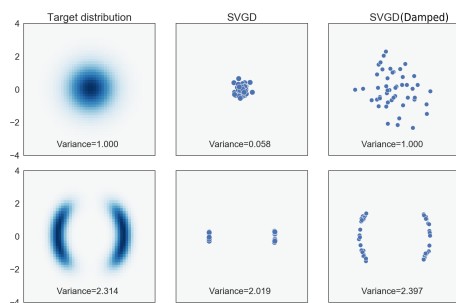

Figure 5: Points represent converged particles of SVGD (middle) and damped SVGD (right). We also report the dimension-averaged variances in the figure. In the Gaussian case (top), SVGD (damped) exactly learns the target variance.

## 6 RELATED WORKS

**Guarantees and Applications of SVGD.** Liu and Wang (2018); Lu et al. (2019) characterized SVGD in the mean-field limit and showed the weak convergence to the target distribution. Under further assumptions on the potential and kernel, quantitative convergence rate can be derived (Duncan et al., 2019; Korba et al., 2020). In addition, Liu and Wang (2018) shows SVGD using kernels with finite-dimensional feature maps, exactly estimates the expectations for some set of functions, casting SVGD as a moment matching method. In the high-dimensional setting, Zhuo et al. (2017); Wang et al. (2018) observed in experiments that particles driven by SVGD tend to underestimate the marginal variance, but did not provide any quantitative understanding of the phenomenon.

On the application side, Haarnoja et al. (2017); Liu et al. (2017) adopt SVGD to learn a stochastic sampling network to approximate the policy in Q-learning (Sutton et al., 1998), whereas in Gangwani et al. (2018), SVGD encourages diverse policies for exploration. SVGD can be used in meta-learning to quickly obtain parameter samples from training sets (Yoon et al., 2018). Recent works also applied SVGD in Batch Bayesian optimization (Gong et al., 2019) and the learning of mixture models (Wang and Liu, 2019). Leveraging the Markov blanket structure, SVGD also achieves strong performance in learning graphical models (Zhuo et al., 2017; Wang et al., 2018).

**Stein's Method.** Stein's method provides powerful tools in approximating probability distributions and specifying convergence rates (Erdogdu, 2016; Gorham et al., 2016). Liu and Wang (2016) utilizes the connection between Stein's operator and the gradient of KL divergence to construct particle inference algorithm. Stein's lemma is also useful in implicit variational inference (Huszár, 2017) to estimate the score using samples from an implicit distribution (Li and Turner, 2017; Shi et al., 2018). Related to our analysis in Section 4, Erdogdu et al. (2016) observed that algorithms that are equivalent in expectation via Stein's lemma might have different convergence properties. In addition, the "curse of dimensionality" of kernel Stein estimators has also been studied in Oates et al. (2016).

## 7 CONCLUSION

We analyzed the variance collapse of SVGD in high dimensions based on a connection between SVGD and a proposed MMD-descent algorithm. We qualitatively identified factors that lead to this phenomenon, and also quantitatively characterized the equilibrium variance in the proportional limit for simple models. Looking forward, we believe that understanding interacting particle systems (SVGD, two-layer neural nets, etc.) in the proportional limit instead of the mean-field limit is an important direction. In addition, while our analysis confirms the variance collapse of the original algorithm, it remains possible that certain dimension reduction schemes could alleviate the issue; Chen and Ghattas (2020); Gorham et al. (2020); Gong et al. (2020) proposed dimension-reduced versions of SVGD, and it would be interesting to theoretically justify these approaches.

---

[7]We however note that due to the distribution-specific derivation of $\lambda$, we should *not* expect the proposed modification to be a general solution to the variance collapse problem of SVGD.

ACKNOWLEDGEMENT

JB was supported by NSERC Grant [2020-06904], CIFAR AI Chairs program, Google Research Scholar Program and Amazon Research Award. MAE was supported by NSERC Grant [2019-06167], Connaught New Researcher Award, CIFAR AI Chairs program, and CIFAR AI Catalyst grant. MG was supported in part by Microsoft Research and a Canadian CIFAR AI Chair held at the Vector Institute. TS was partially supported by JSPS KAKENHI (18H03201), Japan Digital Design and JST CREST. SS was supported by a Connaught New Researcher Award and a Connaught Fellowship.

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

TABLE OF CONTENTS

# A    ADDITIONAL RESULTS

## A.1    ADDITIONAL RESULTS FOR SECTION 5

We include additional figures to illustrate our analytic results in Section 5. In Figure 6 we plot the stationary variance of SVGD and MMD-descent (with the Gaussian RBF kernel and median bandwidth heuristic) under fixed dimensionality $d$ against varying number of particles $n$. As shown in Figure 6(a), the particle variance of SVGD scales linearly with $1/d$ when $\gamma > 1$ (i.e., $d > n$), as predicted by Proposition 3 and Corollary 4; this again confirms the variance collapse phenomenon. When $n > d$, our analysis cannot predict the stationary variance accurately, but the value empirically approaches the target variance from below as $\gamma$ decreases.

**Non-asymptotic Correction for MMD-descent.**    As noted in Section 5, there is a small gap between the empirical particle variance of MMD-descent and the prediction of Proposition 3, due to finite-particle error in the empirical simulation. To characterize this discrepancy, we now consider a regime where $n$ is fixed and $d$ is large. In this case, we show that SVGD collapses all the particles to 0, whereas MMD-descent with the Gaussian RBF kernel can still estimate the marginal variance up to $O(1/n)$ error (see Appendix C for derivation).

**Proposition 7.**    *Consider the setting of fixed $n$ and $d \to \infty$, then given (A2)(A3), particles driven by SVGD (2) collapses to zero variance: $v^{SVGD} \to 0$, whereas MMD-descent (4) with the Gaussian RBF kernel leads to $v^{MMD} \to (n-1)/(n+1)$.*

Figure 6(b) demonstrates the agreement between the proposition and finite-particle behavior of MMD-descent. Observe that the finite-particle error decays at $1/n$ and is independent to the dimensionality. In other words, our analysis suggests that to achieve non-trivial estimation of the target variance, SVGD requires $\Theta(d)$ particles, whereas for MMD-descent $O_d(1)$ particles suffice.

Based on this observation, we speculate that to match the accuracy of MMD-descent with $n$ particles (in terms of marginal variance), SVGD requires $N = nd$ particle. This conjecture is empirically confirmed in Figure 6(c): when we increase the number of particles in SVGD by an additional factor of $d$, then the equilibrium variance on longer depends on the dimensionality and becomes similar to that of MMD-descent with $n$ particles. This again suggests that SVGD may not be reliable unless the number of particles scale with the problem dimensionality.

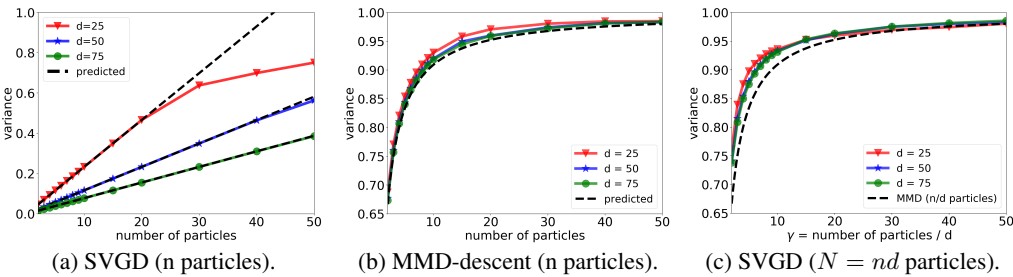

(a) SVGD ($n$ particles).        (b) MMD-descent ($n$ particles).        (c) SVGD ($N = nd$ particles).

Figure 6: Dimension-averaged marginal variance of particles converged under (a) SVGD, (b) MMD and (c) SVGD with $N = nd$ particles. The target is a unit Gaussian, and we employ the Gaussian RBF kernel with median bandwidth. Dashed lines correspond to predicted values. (a) when $n, d$ jointly scales and $d > n$, the variance of SVGD scales linearly with $n$ and $1/d$ as predicted by Proposition 3. (b) for small $n$, variance of MMD-descent approaches 1 as $n$ increases (independent to $d$), which agrees with Proposition 7. (c) if the particle size of SVGD is up-scaled by a factor of $d$, then the underestimation of variance is of order $O(1/n)$.

**Variance Collapse in Almost-Gaussian Target.**    In Section 5 we focus on Gaussian target for precise characterization of the stationary variance. For more general target distributions, if the potential behaves "Gaussian-like" outside of a certain radius (smaller than the true target variance), then under the same assumption on the fixed point particles (A3), we intuitively expect SVGD to underestimate the marginal variance when $\gamma = d/n$ is large.

**Corollary 8.**    *Given $p(\boldsymbol{x}) \propto \exp(-f(x))$ satisfying $\mathbb{E}_p[\boldsymbol{x}] = 0, \mathbb{E}[\boldsymbol{x}\boldsymbol{x}^\top] = \mathbf{I}_d$, and assume $f$ exhibits Gaussian tail growth outside a Euclidean ball with radius 1, then given (A1-3), there exists $\gamma^* = d/n > 1$ such that SVGD underestimates the dimension-averaged variance for $\gamma > \gamma^*$.*

While the proposition above only considers quadratic growth of the potential (similar to Mattingly et al. (2002)), we empirically verify that SVGD underestimates the target variance under anisotropy as well as different tail growth condition. In Figure 7(a) we construct the target distribution to be a mixture of two Gaussians with mean $\boldsymbol{\mu}_1, \boldsymbol{\mu}_2$ such that $\boldsymbol{\mu}_1 + \boldsymbol{\mu}_2 = \mathbf{0}$, $\|\boldsymbol{\mu}_1 - \boldsymbol{\mu}_2\|_2 = 1$, whereas in Figure 7(b) we consider a factorized distribution with a cubic-growth potential: $\log p(\boldsymbol{x}) \propto -\prod_{i=1}^d (\boldsymbol{x}^{(i)})^3/3$. In both cases we observe similar variance collapse phenomenon of SVGD across different choices of kernel and bandwidth.

**Log-inverse Kernel.** We also illustrate the variance collapse phenomenon of SVGD with the *log-inverse* kernel proposed in (Chen et al., 2018b). This heavy-tailed kernel is defined as $k(\boldsymbol{x}, \boldsymbol{x}') = \left(\alpha + \log(1 + \|\boldsymbol{x} - \boldsymbol{x}'\|_2^2/\sigma^2)\right)^{-1}$. One can easily verify that it is a Euclidean distance kernel satisfying assumption (A2), and thus we also expect variance collapse to occur in learning the unit Gaussian target (A3). This is indeed confirmed in Figure 7(c)(d), where we see that the log-inverse kernel with both the median heuristic and fixed bandwidth ($\sigma = 1$) underestimates the marginal variance, and in the case of median bandwidth, we can analytically predict the equilibrium variance for $\gamma > 1$ via Proposition 3.

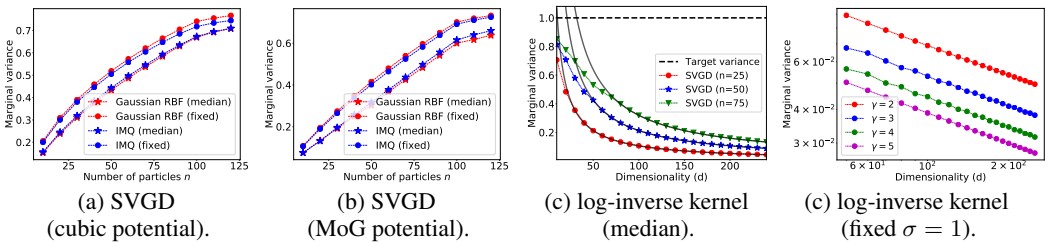

(a) SVGD
(cubic potential).

(b) SVGD
(MoG potential).

(c) log-inverse kernel
(median).

(c) log-inverse kernel
(fixed $\sigma = 1$).

Figure 7: (a)(b) Empirical demonstration of variance collapse beyond Gaussian target. (c)(d) For unit Gaussian target, SVGD with the log-inverse kernel also underestimates the marginal variance.

## A.2 EMPIRICAL VERIFICATION OF (A2)

We empirically validate the near-orthogonality assumption in Section 5 in learning Gaussian targets. In particular, we plot the following quantities for particles driven by SVGD and MMD-descent till convergence (from random initialization):

$$\textbf{(i) } d^{-1/2}\max_i\{\|\boldsymbol{x}_i\|_2^2 - \upsilon d\}; \quad \textbf{(ii) } d^{-1/2}\max_{i,j}\{\boldsymbol{x}_i^\top \boldsymbol{x}_j\}.$$

Note that for **(A2)** to hold, the above quantity should decay with the dimensionality $d$.

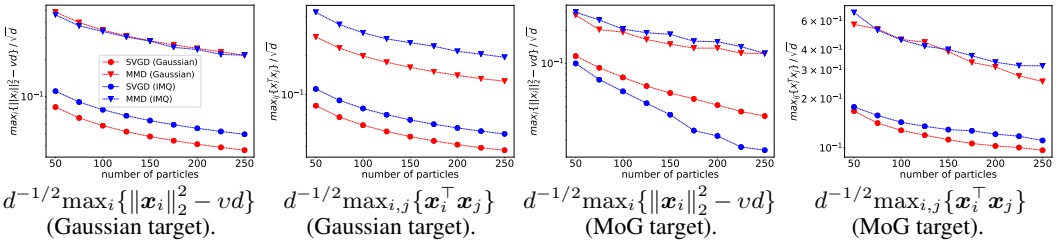

$d^{-1/2}\max_i\{\|\boldsymbol{x}_i\|_2^2 - \upsilon d\}$
(Gaussian target).

$d^{-1/2}\max_{i,j}\{\boldsymbol{x}_i^\top \boldsymbol{x}_j\}$
(Gaussian target).

$d^{-1/2}\max_i\{\|\boldsymbol{x}_i\|_2^2 - \upsilon d\}$
(MoG target).

$d^{-1/2}\max_{i,j}\{\boldsymbol{x}_i^\top \boldsymbol{x}_j\}$
(MoG target).

Figure 8: Quantities of interest in Assumption **(A2)**. We fix $\gamma > 1$ and vary $n, d$ to verify the dependence on $d$. Particles are randomly initialized from $\mathcal{N}(0, 2I_d)$ and optimized till convergence. (a)(b) unit Gaussian target with $\gamma = 2$. (c)(d) MoG target with $\gamma = 3$.

In Figure 8 we validate this hypothesis for both SVGD and MMD-descent on two different kernels: the Gaussian RBF kernel and the IMQ kernel, and two target distributions: unit Gaussian and mixture of Gaussians (MoG). For the Gaussian RBF kernel we use the median bandwidth heuristic, and for the IMQ kernel we set the bandwidth $\sigma = \sqrt{d}$. The MoG model consists of two Gaussians with mean $\boldsymbol{\mu}_1 + \boldsymbol{\mu}_2 = \mathbf{0}$, $\|\boldsymbol{\mu}_1 - \boldsymbol{\mu}_2\|_2 = 1$. For both algorithms we initialize the particles from $\mathcal{N}(0, 2I_d)$. Observe that in all cases the quantities of interest indeed decrease with $d$. We leave the rigorous justification of this observation as future work.

### A.3 ADDITIONAL RESULTS FOR MODIFIED SVGD

In Section 5 we proposed a modification of SVGD based on derivations of learning a unit Gaussian target distribution in high dimensions (and kernels with the median bandwidth),

$$\Delta^{\text{Damp}}(\boldsymbol{x}_i) = \sum_{j \neq i}[S_1(\boldsymbol{x}_j, \boldsymbol{x}_i) + S_2(\boldsymbol{x}_j, \boldsymbol{x}_i)] + \lambda S_1(\boldsymbol{x}_i, \boldsymbol{x}_i),$$

where $\lambda = \max\{1, (f(1) - \gamma^{-1}f'(1))/f(0)\}$. Note that $\lambda = 1$ recovers the original SVGD update, whereas under extreme overparameterization ($p \gg n$) we have $\lambda = f(1)/f(0) < 1$ due to (A3). In addition, larger $\lambda$ implies smaller stationary variance of SVGD, vice versa. In the figures below we refer to the SVGD update with $\lambda_{\min} = \min\{1, (f(1) - \gamma^{-1}f'(1))/f(0)\}$ as "fully-damped", and the update with $\lambda$ in between $\lambda_{\min}$ and 1 as "intermetidate".

**Bayesian Neural Network.** We now apply this modified update to the BNN setting discussed in Section 3 (for detailed setup see Section D). We initialize 100 particles from a standard normal distribution and then optimize them via (modified) SVGD with the Gaussian RBF kernel and median bandwidth for 50000 steps with learning rate $\eta = 5e - 3$. As shown in Figure 9, the original SVGD update (Figure (b), original) significantly underestimates the target variance (estimated using HMC), and as we decrease $\lambda$, the variance of the predictions gradually increases (Figure (c), intermediate). Observe that at our proposed damping factor (Figure (d), fully-damped), the modified update produces diverse samples similar to HMC.

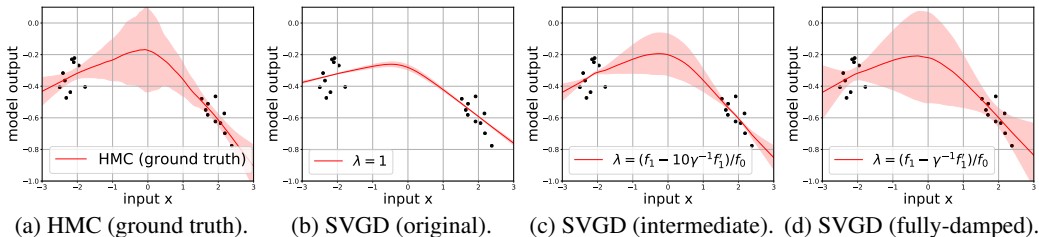

    (a) HMC (ground truth).    (b) SVGD (original).    (c) SVGD (intermediate).    (d) SVGD (fully-damped).

Figure 9: BNN experiment ($\gamma = 100$) for the proposed modification of SVGD (Gaussian RBF kernel). Observe that the original SVGD update (b) significantly underestimates that target variance, whereas the proposed modification (d) leads to more diverse predictions.

In real-world problems, it is not likely that our proposed modification (derived for Gaussian target) always results in optimal performance. This being said, Figure 9 suggests that practitioners could tune the damping term $\lambda$ in the range $[f(1)/f(0), 1]$ and adjust the desired level of uncertainty. Theoretical justification of our modified update is left as future work.

**Bayesian Logistic Regression.** We also conduct additional experiment of the modified (damped) SVGD update in a Bayesian logistic regression problem. Given input $Z = \{\boldsymbol{z}_i\}_{i=1}^m \in \mathbb{R}^{m \times d}$, labels $\boldsymbol{y} = \{y_i\}_{i=1}^m \in \mathbb{R}^m$, and parameter $\theta \in \mathbb{R}^d$, we model the Bernoulli conditional distribution with probability $\Pr(y_i = 1|\boldsymbol{z}_i) = 1/(1 + \exp(-\theta^\top \boldsymbol{z}_i))$. We place an isotropic Gaussian prior on $\theta$; the posterior density is given as

$$p(\theta) \propto \exp\left(\mathbf{y}^\top Z\theta - \sum_{i=1}^m \log(1 + \exp(-\theta^\top \boldsymbol{z}_i)) - \frac{\alpha}{2}\|\theta\|_2^2\right).$$

Following Dalalyan (2014), we sample the coordinates of $\boldsymbol{z}_i$ from a Rademacher distribution and then normalize the vector by its Euclidean norm; the labels are generated from a Bernoulli distribution with true parameters $\theta_* = \mathbf{1}_d$; we set $m = 500, d = 100$, and the regularization parameter $\alpha = 1$.

We report the dimension-averaged marginal variance of the particles (in the $n < d$ regime) optimized by variants of SVGD in Figure 10(b). We use the Gaussian RBF kernel with the median bandwidth, and for the "intermediate" update we set $\lambda_{\min} = \min\{1, (f(1) - 2\gamma^{-1}f'(1))/f(0)\}$. The ground truth variance is estimated by Langevin Monte Carlo (see Appendix A.4 for more discussion). Figure 10(b) illustrates that for this problem, the original SVGD update (red) underestimates the target variance, similar to the Gaussian case. However, our heuristic modification (fully damped,

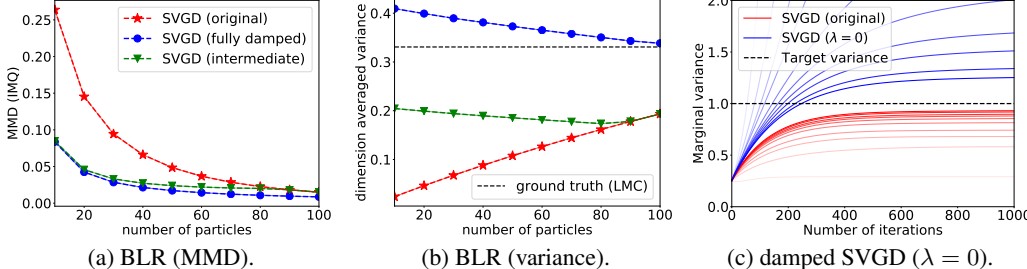

Figure 10: (a)(b) MMD (IMQ kernel) and dimension-averaged variance of SVGD particles in Bayesian logistic regression experiment. Observe that the modified update (blue) leads to smaller MMD (a), but may overestimate the target variance in the small-particle regime (b). (c) Modified SVGD with $\lambda = 0$ leads to diverging particles. Darker color represents larger particle size $n$.

blue) does not remove this bias completely – this is not surprising because the correction term is derived only for the isotropic Gaussian target. In particular, the damped update might overestimate the dimension-averaged variance in the small-particle regime. Nevertheless, we observe that the proposed modification generates more diverse samples (with variance closer to the true value).

To further quantify the convergence of the particles, in Figure 10(a) we report the MMD between the SVGD particles and (approximate) ground-truth samples obtained from LMC; we employ the V-statistics MMD with the IMQ kernel (fixed bandwidth). Observe that the modified update (blue, green) results in smaller MMD compared to the original algorithm (red).

**Complete Removal of $S_1(\boldsymbol{x}_i, \boldsymbol{x}_i)$.** Recall that the SVGD update can be decomposed into three terms: 1) driving force from each particle itself $S_1(\boldsymbol{x}_i, \boldsymbol{x}_i)$; 2) driving force from other particles $\sum_{j \neq i} S_1(\boldsymbol{x}_j, \boldsymbol{x}_i)$; 3) repulsive force $\sum_j S_2(\boldsymbol{x}_j, \boldsymbol{x}_i)$ (note that $S_2(\boldsymbol{x}_i, \boldsymbol{x}_i) = 0$). Specifically, $S_1(\boldsymbol{x}_i, \boldsymbol{x}_i)$ has a large impact on the update and the variance collapse: in the modification above, we are able to generate diverse samples just by decreasing the strength of this single term.

One might speculate that instead of using the specific $\lambda$ derived for Gaussian target, an easier alternative would be to completely remove $S_1(\boldsymbol{x}_i, \boldsymbol{x}_i)$, i.e., setting $\lambda = 0$. This is however not the case. Following the same derivation as in Appendix C, one can see that when $\lambda = 0$, Equation 13 does not equilibrate at any finite variance for $\gamma > 1$. We empirically validate this finding in Figure 10(c), where we fix $d = 50$ and vary the particle size $n$ from 25 to 500, and optimize the particle using (modified) SVGD with the Gaussian RBF kernel and the median bandwidth. Observe that while the original SVGD (red) underestimates the target variance, setting $\lambda = 0$ (blue) leads to significant overestimation or even divergence when $d > n$.

One minor remark is that heuristically speaking, SVGD with $\lambda = 0$ roughly resembles MMD-descent with fixed log derivative $S_1$ (see Figure 2(b) in Section 4), due to the presence of deterministic bias, and that the driving force in both updates involves the log derivative $S_1$, but does not include the "self" term $S_1(\boldsymbol{x}_i, \boldsymbol{x}_i)$. As a result, we observe an overestimation of marginal variance (or diverging particles) in both modified updates.

### A.4 DISCUSSION ON OTHER SAMPLING ALGORITHMS

In addition to SVGD and MMD-descent, there are many other approaches to draw (approximate) samples from a given target distribution. We briefly discuss two popular methods: Langevin Monte Carlo (LMC) and Herding. In particular, we show that LMC does not suffer from variance collapse in high dimensions; in fact, under some conditions, one particle suffices to estimate the dimension-averaged variance of the target.

**Langevin Algorithm.** Langevin Monte Carlo (LMC) is a time discretization of the Langevin diffusion, which can be interpreted as the (Wasserstein) gradient flow of relative entropy (Jordan et al., 1998). Convergence rate of the discrete-time algorithm has been analyzed under various metrics (Dalalyan, 2014; Durmus and Moulines, 2017; Erdogdu et al., 2021) and assumptions on the target distribution (Vempala and Wibisono, 2019; Erdogdu and Hosseinzadeh, 2020).

Here we provide a short sketch that existing quantitative convergence guarantee of LMC implies that the algorithm does not underestimate the dimension-averaged variance. For strongly convex and smooth potentials (which includes the Gaussian setting in Section 5), Erdogdu et al. (2021, Theorem 4) showed that when the strong convexity parameter is dimension-free, then under appropriate choice of step size, running LMC for $T = \tilde{\mathcal{O}}(d)$ iterations leads to $\chi^2(q_T\|p) = \mathcal{O}(1)$. We denote the covariance $\mathbb{E}_p[\boldsymbol{x}\boldsymbol{x}^\top] = \Sigma$, and assume $\mathbb{E}_p[\boldsymbol{x}] = 0, \mathrm{Tr}(\Sigma) = d$ WLOG. By properties of the $\chi^2$-divergence and strongly log-concave concentration, given one sample $\boldsymbol{x}_q$ drawn from $q_T$,

$$\Pr\left(\big|\, \|\boldsymbol{x}_q\|_2 - \mathbb{E}_p\,\|\boldsymbol{x}\|_2 \,\big| > t\right) \lesssim \exp\left(-t^2\right).$$

This concentration directly implies that in the high-dimensional limit $d \to \infty$ (Assumption 1), we have $\|\boldsymbol{x}_q\|_2^2 / \mathrm{Tr}(\Sigma) \xrightarrow{p} 1$. In other words, the dimension-averaged variance $\mathrm{Tr}(\Sigma)/d$ can be reliably estimated using *one single particle* driven by LMC for $\tilde{\mathcal{O}}(d)$ steps. We remark that similar results are expected to hold true under more general conditions such as the *log-Sobolev inequality* on the target distribution, as well as other variants of the algorithm (Roberts and Tweedie, 1996; Cheng et al., 2018b). We therefore speculate that variance collapse is a unique property of certain *interacting* particle algorithms such as SVGD.

**Kernel Herding.** Consider the approximation of an intractable distribution $p(\boldsymbol{x})$ with a set of particles $X = \{\boldsymbol{x}_i\}_{i=1}^n$. To generate these particles, the kernel herding algorithm was introduced by Welling (2009) to minimize the MMD between the particles and the target distribution. The algorithm proceeds in a greedy manner: given the current set of selected particles $\{\boldsymbol{x}_1, \cdots, \boldsymbol{x}_{n-1}\}$, the next particle is chosen based on the following:

$$\boldsymbol{x}_n \leftarrow \underset{x}{\operatorname{argmin}}\ \mathrm{MMD}^2\left(p, \frac{1}{n}\left(\sum_{i=1}^{n-1}\delta_{\boldsymbol{x}_i} + \delta_x\right)\right) = \underset{x}{\operatorname{argmax}}\ \mathbb{E}_{\boldsymbol{y}\sim p}[k(\boldsymbol{x},\boldsymbol{y})] - \frac{1}{n}\sum_{i=1}^{n-1}k(\boldsymbol{x},\boldsymbol{x}_i).$$

Intuitively, the first term encourages sampling in high density areas for the target, whereas the second term discourages drawing samples close to existing ones. It has been shown that the kernel herding algorithm reduces the MMD at a rate $\mathrm{O}(\frac{1}{N})$ for finite-dimensional Hilbert spaces $\mathcal{H}$ (Welling, 2009; Bach et al., 2012; Huszár and Duvenaud, 2012).

Note that the same procedure can also be used to greedily minimize the kernel Stein discrepancy (Chen et al., 2018b; 2019). In addition, the non-greedy gradient descent update on KSD (analogous to MMD-descent) has been recently analyzed (Korba et al., 2021). We leave the high-dimensional characterization of these algorithms as future work.

## A.5 ADDITIONAL FIGURES FOR SECTION 4

In Section 4 we conducted experiments that qualitatively identified the cause of variance collapse, primarily for the Gaussian RBF kernel. Here we reproduce these findings for the IMQ kernel: $k(\boldsymbol{x},\boldsymbol{y}) = 1/\sqrt{1 + \|\boldsymbol{x}-\boldsymbol{y}\|_2^2 / 2\sigma^2}$, which is also commonly-used in SVGD. As shown in Figure 11, results are qualitatively similar to that of Gaussian RBF kernel.

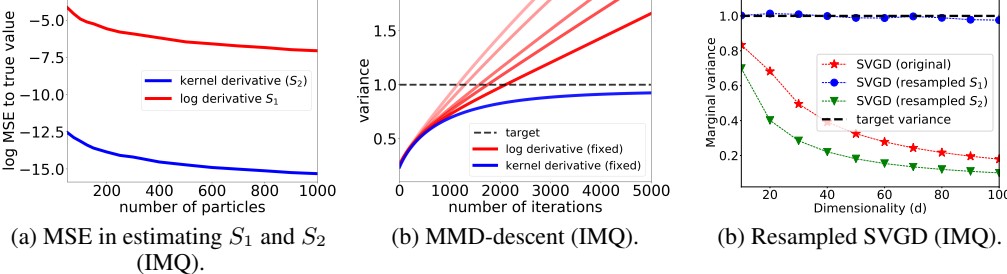

(a) MSE in estimating $S_1$ and $S_2$ (IMQ).

(b) MMD-descent (IMQ).

(b) Resampled SVGD (IMQ).

Figure 11: Learning an isotropic Gaussian using the IMQ kernel. (a) Integration by parts with the IMQ kernel leads to a large discrepancy in the variance of $S_1$ and $S_2$. (b) MMD with IMQ kernel leads to divergence under $\Delta_1^{\mathrm{MMD}}$ (*log derivative*) with fixed target samples. (c) IMQ-SVGD with resampled $S_1$ (blue) correctly estimates the target variance, but redrawing $S_2$ (green) fails to provide more accurate samples.

Finally, we provide additional empirical evidence that SVGD with proper resampling procedure (see algorithm 1) reliably estimates the target variance. We consider the cubic-growth potential: $p(\boldsymbol{x}) \propto \exp\left(-\prod_{i=1}^{d}(\boldsymbol{x}^{(i)})^3/3\right)$, and optimize the particles using SVGD with the Gaussian RBF kernel (fixed bandwidth $\sigma = \sqrt{d}$). As shown in Figure 12, the original SVGD update (red) underestimates the dimension-averaged marginal variance inversely proportional to the problem dimensionality; in contrast, when $S_1$ is resampled (blue), then the estimated variance is stable as $d$ increases.

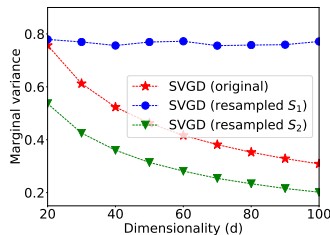

Figure 12: Resampled SVGD in learning the cubic-growth potential defined in Appendix A.1.

## B  DERIVATION OF PROPOSITIONS IN SECTION 4

**Proof of Proposition 2.**  For simplicity we first assume the target is zero-centered: $\boldsymbol{y} \sim \mathcal{N}(0, I_d)$. In this case when the kernel is Gaussian RBF, the expectation of $S_1$, which we denote as $\mu_{\boldsymbol{x}}$, has the following closed-form: $\mu_{\boldsymbol{x}} = -\frac{\sigma^d}{(1+\sigma^2)^{d/2+1}} \exp\left(-\frac{\|\boldsymbol{x}\|_2^2}{2+2\sigma^2}\right) \boldsymbol{x}$. We compute the mean squared error of interest as follow:

$$
\begin{aligned}
\mathrm{MSE}_p[S_1(\boldsymbol{y}, \boldsymbol{x})] &= \int_{\boldsymbol{y}} \|S_1(\boldsymbol{y}, \boldsymbol{x}) - \mu_{\boldsymbol{x}}\|_2^2\, p(\boldsymbol{y}) d\boldsymbol{y} \\
&= \int_{\boldsymbol{y}} \| -\boldsymbol{y} k(\boldsymbol{x}, \boldsymbol{y}) - \mu_{\boldsymbol{x}}\|_2^2\, p(\boldsymbol{y}) d\boldsymbol{y} \\
&= \int_{\boldsymbol{y}} k^2(\boldsymbol{x}, \boldsymbol{y}) \boldsymbol{y}^T \boldsymbol{y}\, p(\boldsymbol{y}) d\boldsymbol{y} + 2\mu_{\boldsymbol{x}}^T \int_{\boldsymbol{y}} k(\boldsymbol{x}, \boldsymbol{y}) \boldsymbol{y}\, p(\boldsymbol{y}) d\boldsymbol{y} + \mu_{\boldsymbol{x}}^T \mu_{\boldsymbol{x}} \int_{\boldsymbol{y}} p(\boldsymbol{y}) d\boldsymbol{y} \\
&= \frac{e^{-\frac{\|\boldsymbol{x}\|_2^2}{2+\sigma^2}} \sigma^d}{(2+\sigma^2)^{d/2+1}} (2\|\boldsymbol{x}\|_2^2 + d\sigma^2) - \frac{2 e^{-\frac{\|\boldsymbol{x}\|_2^2}{1+\sigma^2}} \sigma^{2d}}{(1+\sigma^2)^{d+2}} \|\boldsymbol{x}\|_2^2 + \frac{e^{-\frac{\|\boldsymbol{x}\|_2^2}{1+\sigma^2}} \sigma^{2d}}{(1+\sigma^2)^{d+2}} \|\boldsymbol{x}\|_2^2 \\
&= \|\boldsymbol{x}\|_2^2 \left[ \frac{2 e^{-\frac{\|\boldsymbol{x}\|_2^2}{2+\sigma^2}}}{\sigma^2+2} \left(\frac{\sigma^2}{2+\sigma^2}\right)^{d/2} - \frac{e^{-\frac{\|\boldsymbol{x}\|_2^2}{1+\sigma^2}}}{(\sigma^2+1)^2} \left(\frac{\sigma^2}{1+\sigma^2}\right)^d \right] + d e^{-\frac{\|\boldsymbol{x}\|_2^2}{2+\sigma^2}} \left(\frac{\sigma^2}{2+\sigma^2}\right)^{d/2+1}.
\end{aligned}
$$

Similarly, for the kernel derivative $S_2$ we have,

$$
\begin{aligned}
\mathrm{MSE}_p[S_2(\boldsymbol{y}, \boldsymbol{x})] &= \int_{\boldsymbol{y}} \|S_2(\boldsymbol{y}, \boldsymbol{x}) - (-\mu_{\boldsymbol{x}})\|_2^2\, p(\boldsymbol{y}) d\boldsymbol{y} \\
&= \int_{\boldsymbol{y}} \| \frac{\boldsymbol{x} - \boldsymbol{y}}{\sigma^2} k(\boldsymbol{x}, \boldsymbol{y}) + \mu_{\boldsymbol{x}}\|_2^2\, p(\boldsymbol{y}) d\boldsymbol{y} \\
&= \frac{1}{\sigma^4} \int_{\boldsymbol{y}} k^2(\boldsymbol{x}, \boldsymbol{y}) \boldsymbol{y}^T \boldsymbol{y}\, p(\boldsymbol{y}) d\boldsymbol{y} - \frac{2\boldsymbol{x}}{\sigma^4} \int_{\boldsymbol{y}} k^2(\boldsymbol{x}, \boldsymbol{y}) \boldsymbol{y}\, p(\boldsymbol{y}) d\boldsymbol{y} - \frac{2\mu_{\boldsymbol{x}}}{\sigma^2} \int_{\boldsymbol{y}} k(\boldsymbol{x}, \boldsymbol{y}) \boldsymbol{y}\, p(\boldsymbol{y}) d\boldsymbol{y} \\
&\quad + \frac{\boldsymbol{x}^T \boldsymbol{x}}{\sigma^4} \int_{\boldsymbol{y}} k^2(\boldsymbol{x}, \boldsymbol{y})\, p(\boldsymbol{y}) d\boldsymbol{y} + \frac{2\boldsymbol{x}^T \mu_{\boldsymbol{x}}}{\sigma^2} \int_{\boldsymbol{y}} k(\boldsymbol{x}, \boldsymbol{y})\, p(\boldsymbol{y}) d\boldsymbol{y} + \mu_{\boldsymbol{x}}^T \mu_{\boldsymbol{x}} \int_{\boldsymbol{y}} p(\boldsymbol{y}) d\boldsymbol{y} \\
&= \frac{e^{-\frac{\|\boldsymbol{x}\|_2^2}{2+\sigma^2}} \sigma^{d-4}}{(2+\sigma^2)^{d/2+1}} (2\|\boldsymbol{x}\|_2^2 + d\sigma^2) - \frac{4 e^{-\frac{\|\boldsymbol{x}\|_2^2}{2+\sigma^2}} \sigma^{d-4}}{(2+\sigma^2)^{d/2+1}} \|\boldsymbol{x}\|_2^2 + \frac{2 e^{-\frac{\|\boldsymbol{x}\|_2^2}{1+\sigma^2}} \sigma^{2d-2}}{(1+\sigma^2)^{d+2}} \|\boldsymbol{x}\|_2^2 \\
&\quad + \frac{e^{-\frac{\|\boldsymbol{x}\|_2^2}{2+\sigma^2}} \sigma^{d-4}}{(2+\sigma^2)^{d/2+1}} (2+\sigma^2) \|\boldsymbol{x}\|_2^2 - \frac{2 e^{-\frac{\|\boldsymbol{x}\|_2^2}{1+\sigma^2}} \sigma^{2d-2}}{(1+\sigma^2)^{d+1}} \|\boldsymbol{x}\|_2^2 + \frac{e^{-\frac{\|\boldsymbol{x}\|_2^2}{1+\sigma^2}} \sigma^{2d}}{(1+\sigma^2)^{d+2}} \|\boldsymbol{x}\|_2^2 \\
&= \frac{e^{-\frac{\|\boldsymbol{x}\|_2^2}{2+2\sigma^2}}}{\sigma^4} \left(\frac{\sigma^2}{2+\sigma^2}\right)^{d/2+1} (d + \|\boldsymbol{x}\|_2^2) + \frac{3 e^{-\frac{\|\boldsymbol{x}\|_2^2}{1+\sigma^2}}}{(1+\sigma^2)^2} \left(\frac{\sigma^2}{1+\sigma^2}\right)^d \|\boldsymbol{x}\|_2^2.
\end{aligned}
$$

The simplification above largely follows from $\mathbb{E}_{\boldsymbol{x} \sim \mathcal{N}(\mu, \Sigma)}[\|\boldsymbol{x}\|_2^2] = \mu^T \mu + \text{Tr}(\Sigma)$. For non-centered $\boldsymbol{y} \sim \mathcal{N}(\boldsymbol{a}, I_d)$, note that replacing $\|\boldsymbol{x}\|_2^2$ with $\|\boldsymbol{x} - \boldsymbol{a}\|_2^2$ does not affect the order dependence on $d$ as long as $\|\boldsymbol{a}\|_2^2 = O(d)$, and therefore the order that we aim to estimate remains the same.

Given the bandwidth heuristic $\sigma = \Theta(\sqrt{d})$ and $\|\boldsymbol{x}\|_2^2 = d$, one can easily verify that:
$$\text{MSE}_p[S_2(\boldsymbol{y}, \boldsymbol{x})] \in \Theta(d^{-1}), \quad \text{MSE}_p[S_1(\boldsymbol{y}, \boldsymbol{x})] \in \Theta(d).$$

To extend the result to general Euclidean distance kernel $k(\boldsymbol{x}, \boldsymbol{y}) = f\left(\frac{\|\boldsymbol{x} - \boldsymbol{y}\|_2^2}{2\sigma^2}\right)$ and strong log-concave distributions satisfying $\left\|\mathbb{E}_{p(\boldsymbol{y})}[\boldsymbol{y}]\right\|_2^2 = O(d)$ and $\mathbb{E}_{p(\boldsymbol{y})}[\|\boldsymbol{y}\|_2^2] = O(d)$, first note that

$$\mathbb{E}_{\boldsymbol{y} \sim p}[\|\mu_{\boldsymbol{x}}\|_2] = \mathbb{E}_{\boldsymbol{y} \sim p}[\|\nabla_{\boldsymbol{y}} k(\boldsymbol{y}, \boldsymbol{x})\|_2]$$
$$= C_1 \mathbb{E}_{\boldsymbol{y} \sim p}\left[\left\|\frac{\boldsymbol{x} - \boldsymbol{y}}{\sigma^2} \nabla_z f(z)\right\|_2\right], \quad z = \frac{\|\boldsymbol{x} - \boldsymbol{y}\|_2^2}{2\sigma^2}$$
$$\overset{(i)}{\leq} C_2 O(d^{-1})\sqrt{\mathbb{E}_{\boldsymbol{y} \sim p}\left[\|\boldsymbol{x} - \boldsymbol{y}\|^2\right]} = O(d^{-1/2}),$$

where (i) is by Cauchy-Schwarz and the Lipschitzity of $f$. Similarly, following the expansion of $\text{MSE}_p[S_2]$ and the assumptions that k is upper-bounded, one can verify that $\text{MSE}_p[S_2(\boldsymbol{y}, \boldsymbol{x})] = O(1)$. Finally, we lower-bound the variance of $S_1$ via the following calculation,

$$\mathbb{E}_{\boldsymbol{y} \sim p}\left[\|S_1(\boldsymbol{y}, \boldsymbol{x}) - \mu_{\boldsymbol{x}}\|_2^2\right] = \mathbb{E}_{\boldsymbol{y} \sim p}\left[\|\nabla_{\boldsymbol{y}} \log p(\boldsymbol{y})\|_2^2 k^2(\boldsymbol{x}, \boldsymbol{y})\right] - \|\mu_{\boldsymbol{x}}\|_2^2$$
$$\geq C_1 \mathbb{E}_{\boldsymbol{y} \sim p}\left[\|\boldsymbol{y}\|_2^2 k^2(\boldsymbol{x}, \boldsymbol{y})\right] - \|\mu_{\boldsymbol{x}}\|_2^2$$
$$\overset{(i)}{\geq} C_2 \mathbb{E}_{\boldsymbol{y} \sim \mathcal{N}(0, I)}\left[\|\boldsymbol{y}\|_2^2 e^{-\frac{\|\boldsymbol{x} - \boldsymbol{y}\|_2^2}{\sigma^2}}\right] - \|\mu_{\boldsymbol{x}}\|_2^2 \overset{(ii)}{=} \Omega(d),$$

where (i) is by the strongly log-concavity of $p$ and the fact that the kernel being lower-bounded by a scaled Gaussian RBF kernel, and (ii) directly follows from result in the Gaussian case. Combining the calculations yields the desired statement.

## C  DERIVATION OF PROPOSITIONS IN SECTION 5

In this section we aim to calculate the equilibrium variance of SVGD and MMD-Descent under the proportional asymptotics. We first restate and comment on the Assumptions in Section 5.

- **(A2) Near-orthogonality.** Particles at fixed point of SVGD (or MMD-descent) $\{\boldsymbol{x}_i\}_{i=1}^n$ satisfy $|\boldsymbol{x}_i^\top \boldsymbol{x}_i - dv| < v\epsilon_d$, $|\boldsymbol{x}_i^\top \boldsymbol{x}_j| < v\epsilon_d$ for all $i \neq j$, some $v > 0$, and $d^{-1/2}\epsilon_d \to 0$ as $d \to \infty$ with probability 1.

- **(A4) Gaussian Target.** $p(\boldsymbol{x}) \propto \exp\left(-\frac{1}{2}\boldsymbol{x}^\top \boldsymbol{x}\right)$.

Under (A4) the goal of SVGD or MMD-descent is to draw samples from a unit Gaussian distribution, and we aim to calculate the dimension-averaged variance $v$ of the particles evolved by the two updates. Note that since both SVGD and MMD-descent form an interacting particle system, one can no longer treat the converged particles as i.i.d Gaussian samples, i.e. $\boldsymbol{x}_i \sim \mathcal{N}(\boldsymbol{0}, vI)$. This significantly complicates the analysis in the proportional limit.

However, empirical results (see Figure 8) demonstrate that starting from non-degenerate random initialization, the equilibrium solution of both algorithms admits thin-shell concentration around the radius, i.e., the norm of $\boldsymbol{x}_i$ concentrates around $\sqrt{dv}$, and particles are almost orthogonal to one another. This observation is captured by (A2). We conjecture that such property holds for the updates we consider due to the *repulsive force* pushing the particles away from one another, and thus leading to this near-orthogonal configuration.

Under assumptions (A2)(A4), we are able to compute the stationary variance of both SVGD and MMD-descent in the asymptotic limit. We first provide a detailed derivation for the case of Gaussian RBF kernel, and then extend the analysis to more general settings.

### C.1 GAUSSIAN KERNEL WITH THE MEDIAN BANDWIDTH

**Stationary Variance of SVGD in Proportional Limit.** We aim to solve the stationary point of SVGD update (2). Under (A2) and the median heuristic $\sigma = \sqrt{\text{Med}\{\|x_i - x_j\|_2^2\}/2}$, the fixed point condition for the Gaussian RBF kernel is given as

$$\Delta(x_k) = \frac{1}{n} \sum_{i=1}^{n} \left[ -k(x_i, x_k)x_i + \frac{1}{dv}k(x_i, x_k)(x_k - x_i) \right] = \mathbf{0}, \tag{6}$$

for all $k$, or equivalently

$$\sum_{i=1}^{n} k(x_i, x_k)x_i = \frac{1}{dv+1} \sum_{i=1}^{n} k(x_i, x_k) \cdot x_k. \tag{7}$$

Note that for the left hand side of (7), we have the following equivalence,

$$\text{LHS} = \sum_{i=1}^{n} k(x_i, x_k)x_i = Xk_k,$$

where $X = [x_1, \cdots, x_n] \in \mathbb{R}^{d \times n}$ is the data matrix, $k_k = [k(x_1, x_k), \cdots, k(x_n, x_k)]^\top \in \mathbb{R}^n$ is the $k$-th column of kernel Gram matrix $K \in \mathbb{R}^{n \times n}$. As for the RHS of Equation (7), note that for $i \neq k$, (A2) allows us to take the following Taylor expansion on entries of the kernel matrix around its concentrated value, i.e.,

$$k(x_i, x_k) = \exp\left(-\frac{\|x_i - x_k\|_2^2}{2dv}\right) = e^{-1} + O(\epsilon).$$

Where $\epsilon \cdot d^{1/2} \to 0$. Similarly for $i = k$ we have $k(x_k, x_k) \approx 1$. Combining the equations above,

$$\text{RHS} = \frac{1}{dv+1} \sum_{i=1}^{n} k(x_i, x_k) \cdot x_k = \left(\frac{n+e-1}{(dv+1)e} + O(\epsilon)\right)x_k.$$

Equating the RHS and LHS of Eq (7) in matrix form (over all $k$) we have

$$X \cdot K = \frac{n+e-1}{(dv+1)e}X + X \cdot \text{diag}(\epsilon).$$

where $\text{diag}(\epsilon)$ is a squared matrix where the $i$-th diagonal element is the error from Taylor expansion $\epsilon_i = O(\epsilon)$. Define

$$m = \frac{n+e-1}{dv+1}.$$

The fixed point of SVGD thus simplifies to

$$X \cdot (K - mI_n - \text{diag}(\epsilon)) = 0. \tag{8}$$

Denote $A = K - mI_n - \text{diag}(\epsilon)$, Recall that the $K$ is an Euclidean kernel matrix with $K_{ij} = k(x_i, x_k) = \exp\left(-(2dv)^{-1}\|x_i - x_k\|_2^2\right)$. From Theorem 4 in Bordenave et al. (2013), it follows that the empirical spectrum of $A$, which we write as $\mu(A) = n^{-1}\sum_{i=1}^{n} \delta_{\lambda_i(A)}$, converges weakly to the following quantity,

$$\mu(A) \to \left(1 - \frac{2}{e} - m\right) + \frac{1}{e}\mu\left(\frac{1}{dv}X^\top X\right) + \mu(\text{diag}(\epsilon)),$$

In addition, define $S = n/(n-1)I_n - 1/(n-1)\mathbf{1}_n\mathbf{1}_n^\top$, then by the Hoffman-Wielandt inequality (see Bordenave et al. (2013, Lemma 6)) we have

$$W_2\left(\mu\left(\frac{1}{dv}X^\top X\right), \mu(S)\right) \leq \sqrt{\frac{1}{n}\text{tr}\left(\frac{1}{dv}X^\top X - S\right)^2} = \sqrt{n^{-1} \cdot (nO(\epsilon))^2} \to 0,$$

where we used (A2), and $W_2(\cdot, \cdot)$ is the 2-Wasserstein distance. Hence, we know that

$$
\mu(A) \to \left(1 - \frac{2}{e} - m\right) + \frac{1}{e}\mu\left(\frac{1}{dv}X^\top X\right) + \mu(\operatorname{diag}(\boldsymbol{\epsilon}))
$$

$$
\to \left(1 - \frac{2}{e} - m\right) + \frac{1}{e}\mu(S) + \mu(\operatorname{diag}(\boldsymbol{\epsilon})). \tag{9}
$$

When $\gamma > 1$ (i.e. $d > n$), Equation (8) requires $\mu(A) \to 0$. Consequently,

$$
1 - \frac{2}{e} + \frac{n}{e(n-1)} - m = 0 \quad \Leftrightarrow \quad m \to 1 - e^{-1}.
$$

Therefore, from the definition of $m$ we have the desired result.

$$
v^{\mathrm{SVGD}} \to \frac{n}{d(e-1)} = \frac{1}{e-1}\frac{1}{\gamma}. \tag{10}
$$

**Stationary Variance of MMD-descent in Proportional Limit.** First note that for the Gaussian RBF kernel, the driving force of MMD-descent (4) admits the following closed-form,

$$
\mathbb{E}_{\boldsymbol{y}\sim p}[S_2(\boldsymbol{y}, \boldsymbol{x})] = -\frac{\sigma^d}{(1+\sigma^2)^{d/2+1}}\exp\left(-\frac{\|\boldsymbol{x}\|_2^2}{2+2\sigma^2}\right)\boldsymbol{x}.
$$

This can be verified via simple numerical calculation:

$$
\int p(\boldsymbol{y})k(\boldsymbol{x}, \boldsymbol{y})\nabla_{\boldsymbol{y}}\log p(\boldsymbol{y})d\boldsymbol{y}
$$

$$
= \int e^{-\frac{\|\boldsymbol{x}-\boldsymbol{y}\|_2^2}{2\sigma^2}}(-\boldsymbol{y})\frac{1}{\sqrt{(2\pi)^d}}e^{-\frac{\boldsymbol{y}^\top \boldsymbol{y}}{2}}d\boldsymbol{y}
$$

$$
= -\frac{1}{\sqrt{(2\pi)^d}}\int_{\boldsymbol{y}}\boldsymbol{y}e^{-\frac{\left(\frac{1}{\sqrt{1+\sigma^2}}\boldsymbol{x}-\sqrt{1+\sigma^2}\boldsymbol{y}\right)^2}{2\sigma^2}}e^{-\frac{\boldsymbol{x}^\top \boldsymbol{x}}{2+2\sigma^2}}d\boldsymbol{y}
$$

$$
= -\frac{\sigma^d}{(1+\sigma^2)^{d/2+1}}e^{-\frac{\|\boldsymbol{x}\|_2^2}{2+2\sigma^2}}\boldsymbol{x}.
$$

Therefore, the stationary point of MMD-descent satisfies

$$
\Delta\boldsymbol{x}_k = -\frac{\sigma^d}{(1+\sigma^2)^{d/2+1}}e^{-\frac{\|\boldsymbol{x}_k\|_2^2}{2+2\sigma^2}}\boldsymbol{x}_k + \frac{1}{n\sigma^2}\sum_{i\neq k}k(\boldsymbol{x}_k, \boldsymbol{x}_i)(\boldsymbol{x}_k - \boldsymbol{x}_i) = 0,
$$

$\forall k$, or equivalently,

$$
\sum_{i=1}^n k(\boldsymbol{x}_k, \boldsymbol{x}_i)\boldsymbol{x}_k - \left(\frac{dv}{1+dv}\right)^{d/2+1}e^{-\frac{\|\boldsymbol{x}_k\|_2^2}{2+2dv}}n\boldsymbol{x}_k = \sum_{i=1}^n k(\boldsymbol{x}_k, \boldsymbol{x}_i)\boldsymbol{x}_i.
$$

Under assumptions (A2)(A3), similar to the SVGD case, we have the matrix form of the fixed point,

$$
\left[1 + e^{-1}(n-1) - \left(\frac{dv}{1+dv}\right)^{d/2+1}e^{-\frac{dv}{2+2dv}}n\right]X + X\operatorname{diag}(\boldsymbol{\epsilon}) = XK.
$$

with $\epsilon = o(1)$. Following an anlogous calculation, as $n, d \to \infty$ with $d/n = \gamma \in (1, \infty)$ we obtain,

$$
1 + e^{-1}(n-1) - \left(\frac{dv}{1+dv}\right)^{d/2+1}e^{-\frac{dv}{2+2dv}}n \to 1 - \frac{1}{e}. \tag{11}
$$

Moreover, observe that $\lim_{d\to\infty}(dv/(1+dv))^{d/2+1} = e^{-1/(2v)}$. We arrive at the desired result,

$$
v^{\mathrm{MMD}} \to 1.
$$

Following the exact same reasoning, one can show that for fixed bandwidth $\sigma = c\sqrt{d}$ for some constant $c \in \Theta(1)$, MMD-descent also estimates the variance correctly, i.e., $v^{\mathrm{MMD}} = 1$.

**SVGD and MMD-descent with Finite $n$ and Large $d$.** From (A2) we have the following decomposition for $i \neq j$,

$$\|\boldsymbol{x}_i - \boldsymbol{x}_j\|_2^2 = \|\boldsymbol{x}_i\|_2^2 + \|\boldsymbol{x}_j\|_2^2 - 2\boldsymbol{x}_i^\top \boldsymbol{x}_j = 2dv + 2(n-1)^{-1}dv + O(\epsilon) = \frac{2n}{n-1}dv + O(d\epsilon).$$

For finite $n$, this indicates that the Euclidean distance between two particles concentrates around $2(n-1)^{-1}ndv$ (instead of $2dv$); this is to say, the Gaussian RBF kernel admits the the following Taylor expansion for $i \neq j$,

$$k(\boldsymbol{x}_i, \boldsymbol{x}_j) = \exp\left(-\frac{\|\boldsymbol{x}_i - \boldsymbol{x}_k\|_2^2}{2dv}\right) = e^{-\frac{n}{n-1}} + O(\epsilon).$$

Therefore, for the MMD-descent algorithm, (11) reduces to

$$1 + e^{-\frac{n}{n-1}}(n-1) - \left(\frac{dv}{1+dv}\right)^{d/2+1} e^{-\frac{dv}{2+2dv}}n \rightarrow 1 - e^{-\frac{n}{n-1}}.$$

Hence the converged particles of MMD-descent with finite $n$ and infinite $d$ has variance

$$v^{\text{MMD}} \rightarrow \frac{n-1}{n+1}.$$

And for SVGD, one can use the exact same argument to obtain that

$$v^{\text{SVGD}} \rightarrow 0.$$

Note that this result agrees with our characterization in the proportional asymptotic limit by taking $\gamma \rightarrow \infty$ (i.e., $d \gg n$).

## C.2   GENERAL KERNEL WITH ADAPTIVE BANDWIDTH (MEDIAN HEURISTIC)

Recall the definition of the (bandwidth-adjusted) Euclidean distance kernel: $k(\boldsymbol{x}, \boldsymbol{y}) = f\left(\frac{\|\boldsymbol{x}-\boldsymbol{y}\|_2^2}{2\sigma^2}\right)$. Under (A2) and the median heuristic $\sigma = \sqrt{\text{Med}\{\|\boldsymbol{x}_i - \boldsymbol{x}_j\|_2^2\}/2}$, we know that the kernel can be equivalently written as $k(\boldsymbol{x}, \boldsymbol{y}) = f\left(\frac{\|\boldsymbol{x}-\boldsymbol{y}\|_2^2}{2dv}\right)$. Following the same procedure, we have the following fixed point condition,

$$\Delta(\boldsymbol{x}_k) = \frac{1}{n}\sum_{i=1}^{n}\left[-f\left(\frac{\|\boldsymbol{x}_i - \boldsymbol{x}_k\|_2^2}{2dv}\right)\boldsymbol{x}_i - \frac{1}{dv}f'\left(\frac{\|\boldsymbol{x}_i - \boldsymbol{x}_k\|_2^2}{2dv}\right)(\boldsymbol{x}_k - \boldsymbol{x}_i)\right] = \boldsymbol{0}, \qquad (12)$$

for all $k$, or equivalently

$$\sum_{i=1}^{n}\left[f\left(\frac{\|\boldsymbol{x}_i - \boldsymbol{x}_k\|_2^2}{2dv}\right) - \frac{1}{dv}f'\left(\frac{\|\boldsymbol{x}_i - \boldsymbol{x}_k\|_2^2}{2dv}\right)\right]\boldsymbol{x}_i = -\frac{1}{dv}\sum_{i=1}^{n}f'\left(\frac{\|\boldsymbol{x}_i - \boldsymbol{x}_k\|_2^2}{2dv}\right)\boldsymbol{x}_k.$$

Similar to the previous calculation on the Gaussian RBF kernel, applying (A2) and the differentiability of $f$ around 1, we Taylor-expand the kernel and obtain the following equivalence:

$$X \cdot (K - \frac{1}{dv}K') = -\frac{f(0) + (n-1)f'(1)}{dv} \cdot X + X \cdot \text{diag}(\boldsymbol{\epsilon}).$$

where $K$ is the Gram matrix of the kernel $k$, $K'$ is the Gram matrix of its derivative with $K'_{ij} = f'\left(\frac{\|\boldsymbol{x}_i - \boldsymbol{x}_j\|_2^2}{2dv}\right)$. Apply Theorem 4 of Bordenave et al. (2013) to $K$ and $K'$, we have the following relation under (A1)(A2),

$$f(0) - f(1) - \frac{1}{dv}(f'(0) - f'(1)) = -\frac{f(0) + (n-1)f'(1)}{dv}, \qquad (13)$$

which gives the expression of the equilibrium variance,

$$v^{\text{SVGD}} \rightarrow \frac{f'(1)}{f(1) - f(0)} \cdot \frac{1}{\gamma}.$$

**Heuristic Derivation of the Modified (Damped) Update.**  We provide a brief sketch of the modified algorithm in Section 5 for the median bandwidth. Recall that our proposed update introduces a damping term $\lambda f(0) \boldsymbol{x}_k$ in the driving force $S_1$. To derive the optimal $\lambda$ for unit Gaussian target, we incorporate this damping term into the fixed point Equation (12), which leads to the following modification of Equation (13):

$$f(0) - f(1) - \frac{1}{dv}(f'(0) - f'(1)) = (1 - \lambda)f(0) - \frac{f(0) + (n-1)f'(1)}{dv}.$$

Recall the true target variance $v = 1$; this gives following choice of $\lambda$:

$$\lambda = f(0)^{-1} \cdot \left( f(1) - \gamma^{-1} f'(1) \right). \tag{14}$$

Since our goal is to "weaken" the driving force $S_1(\boldsymbol{x}_k, \boldsymbol{x}_k)$, we take the minimum between the derived value in (14) and the default $\lambda = 1$.

### C.3  GENERAL KERNEL WITH FIXED BANDWIDTH

We now consider Euclidean distance kernel with invariant (fixed) bandwidth that scales with the dimensionality $d$, which we write as $k(\boldsymbol{x}, \boldsymbol{y}) = f(\|\boldsymbol{x} - \boldsymbol{y}\|_2^2 / 2d)$. Following (12), we have the following equilibrium condition for the particles,

$$\sum_{i=1}^{n} \left[ f\left( \|\boldsymbol{x}_i - \boldsymbol{x}_k\|_2^2 / 2d \right) - \frac{1}{d} f'\left( \|\boldsymbol{x}_i - \boldsymbol{x}_k\|_2^2 / 2d \right) \right] \boldsymbol{x}_i = -\frac{1}{d} \sum_{i=1}^{n} f'\left( \|\boldsymbol{x}_i - \boldsymbol{x}_k\|_2^2 / 2d \right) \boldsymbol{x}_k.$$

Under Taylor expansion around $v$, the fixed point condition entails that the stationary variance satisfies,

$$f(0) - f(v) + \frac{1}{d}(f'(0) - f'(v)) = -\frac{1}{d}(f(0) + (n-1)f'(v)).$$

$$\iff \quad \frac{f'(v)}{f(v) - f(0)} = \gamma. \tag{15}$$

Note that given (A3) and monotone $f'$, the numerator of the RHS of (15) takes a negative value that increases with $v$, whereas the denominator is also negative but decreases with $v$. This implies that when $\gamma$ becomes larger, $v$ needs to decay towards 0 in order to satisfy the equation.

For the Gaussian RBF kernel with fixed bandwidth $\sigma = \sqrt{d}$, the equation can be easily solved as,

$$v^{\text{RBF}} = \log\left( 1 + \frac{1}{\gamma} \right) < 1,$$

which is a decreasing function of $\gamma > 1$.

On the other hand, for the IMQ kernel with fixed bandwidth $\sigma = \sqrt{d}$, standard calculation yields,

$$v^{\text{IMQ}} = \frac{1}{6} \left( \frac{4\gamma(\gamma + 3)}{\left( 3\gamma^4 \sqrt{3(8\gamma + 27)} + 8\gamma^6 + 36\gamma^5 + 27\gamma^4 \right)^{1/3}} \right.$$

$$\left. + \frac{\left( 3\gamma^4 \sqrt{3(8\gamma + 27)} + 8\gamma^6 + 36\gamma^5 + 27\gamma^4 \right)^{1/3}}{\gamma^2} \right) - \frac{2}{3}.$$

One can numerically verify that the value is less than 1 for $\gamma > 1$ and also non-increasing.

**IMQ Kernel with Dimension-independent Bandwidth.**  Finally, we note that in the context of kernel Stein discrepancy (KSD), the IMQ kernel is often employed without the dimension-dependent bandwidth (Gorham and Mackey, 2017). In this case we write the kernel as $k(\boldsymbol{x}, \boldsymbol{y}) = f(\|\boldsymbol{x} - \boldsymbol{y}\|_2^2 / 2)$, which gives the following stationary condition,

$$\sum_{i=1}^{n} \left[ f\left( \|\boldsymbol{x}_i - \boldsymbol{x}_k\|_2^2 / 2 \right) - f'\left( \|\boldsymbol{x}_i - \boldsymbol{x}_k\|_2^2 / 2 \right) \right] \boldsymbol{x}_i = -\sum_{i=1}^{n} f'\left( \|\boldsymbol{x}_i - \boldsymbol{x}_k\|_2^2 / 2 \right) \boldsymbol{x}_k.$$

Taking Taylor expansion around $dv$ gives

$$f(0) - f(vd) - f'(0) + f'(vd) = -f(0) - (n-1)f'(vd).$$

Note that the LHS is $\Theta(1)$ by (A2); in order for the inequality to hold asymptotically, we need to have $f'(vd) = \Theta(n^{-1})$. On the other hand, for the fixed-bandwidth IMQ kernel,

$$f(a) = (1+a)^{-1/2}; \quad 2f'(a) = -(1+a)^{-3/2}.$$

This implies that as $d$ increases, the stationary variance decays to 0 at a rate of $v = \Theta(d^{-1/3})$. Following the exact same procedure, one can show that the log-inverse kernel with fixed dimension-independent bandwidth also asymptotically collapses the variance of SVGD particles to 0 when $\gamma > 1$; we omit the derivation.

## C.4 "ALMOST-GAUSSIAN" TARGET

For general $p(\boldsymbol{x}) \propto \exp(-f(\boldsymbol{x}))$, the fixed point equation of $\boldsymbol{x}_k$ is given as,

$$\Delta(\boldsymbol{x}_k) = \frac{1}{n} \sum_{i=1}^{n} \left[ -f\left( \frac{\|\boldsymbol{x}_i - \boldsymbol{x}_k\|_2^2}{2\sigma^2} \right) \nabla f(\boldsymbol{x}_i) - \frac{1}{\sigma^2} f'\left( \frac{\|\boldsymbol{x}_i - \boldsymbol{x}_k\|_2^2}{2\sigma^2} \right) (\boldsymbol{x}_k - \boldsymbol{x}_i) \right] = \mathbf{0}.$$

Assume that SVGD does not underestimate that marginal variance, then by (A2) and the quadratic growth, we know that there exists some $\alpha > 0$ such that for every particle $\boldsymbol{x}_k$ and coordinate $m \in [d]$,

$$\left[ \sum_{i=1}^{n} \alpha f\left( \frac{\|\boldsymbol{x}_i - \boldsymbol{x}_k\|_2^2}{2\sigma^2} \right) \boldsymbol{x}_i - \frac{1}{\sigma^2} f'\left( \frac{\|\boldsymbol{x}_i - \boldsymbol{x}_k\|_2^2}{2\sigma^2} \right) \boldsymbol{x}_i \right]_m = -\frac{1}{\sigma^2} \left[ \sum_{i=1}^{n} f'\left( \frac{\|\boldsymbol{x}_i - \boldsymbol{x}_k\|_2^2}{2\sigma^2} \right) \boldsymbol{x}_k \right]_m.$$

For kernels with the median bandwidth, followings the same simplification as in (13), we get

$$X \cdot \left( \alpha K - \frac{1}{dv} K' \right) = -\frac{f(0) + (n-1)f'(1)}{dv} \cdot X + X \cdot \mathrm{diag}(\boldsymbol{\epsilon}).$$

Solving the inequality yields,

$$v = \frac{\alpha}{\gamma} \cdot \frac{f'(1)}{f(1) - f(0)}.$$

Therefore, given any dimension-independent growth of the potential $\alpha$, there exists a large enough $\gamma$ such that $v < 1$, i.e., SVGD underestimates the marginal variance. The case for fixed bandwidth follows from the same line of reasoning, the details of which we omit.

## D EXPERIMENT SETUP

**Bayesian Neural Network.** We consider a BNN with two hidden layers of 100 units. In each layer, the preactivations $\mathbf{s}_{t+1}$ are computed via $\mathbf{s}_{t+1} = (\boldsymbol{W}_t \mathbf{a}_t + \mathbf{b}_t)/\sqrt{h_t + 1}$, where $\mathbf{a}_t \in \mathbb{R}^{h_t}$ are the input activations. The target function $f$ is a BNN with the same architecture, whose weights and biases are randomly generated from standard normal distributions. For the training set, we sampled 10 input locations uniformly from $[-2.5, -1.5]$ and $[1.5, 2.5]$, respectively. We add random noises to the observations, $y = f(\boldsymbol{x}) + \epsilon, \epsilon \sim \mathcal{N}(0, 0.01)$.

We first adopt the Hamiltonian Monte Carlo (HMC) (Neal et al., 2011) to generate asymptotic posterior samples. HMC used 50 independent chains and each chain selected 50 particles with the frequency of 100 iterations after $5k$ burn-in iterations. HMC generates diverse particles as shown in Figure 1(a). We then simulate SVGD and MMD-descent dynamics on this problem. For both updates we use the Gaussian RBF kernel with the median bandwidth heuristic. For SVGD, we evolve 100 particles for $50k$ iterations using learning rate $\eta = 5e - 3$, whereas for MMD-descent, we evolve 10 particles using HMC particles as approximate target samples to compute the driving force term. The particles are either initialized from the (approximate) target distribution (as in Figure 1), or from standard normal distribution (as in Figure 9). Note that the plots do not include observation variance.

**Hyperparameter Setting in Section 5.** For all experiments in Section 5, we initialize the particles from $\mathcal{N}(0, 0.8I_d)$, and run SVGD (or MMD-descent) with learning rate $\eta = 10^{-1}$ for $20k$ iterations. For experiments in the proportional limit (Figure 4(a)(b)), we fix $n = 50$ and vary $d$. For the median heuristic (following Garreau et al. (2017)), we compute the median of Euclidean distance between all particles at each iteration.

