# OpenReview forum: "Understanding the Variance Collapse of SVGD in High Dimensions"
_ICLR.cc/2022/Conference — ICLR 2022 Poster_

### Official Review · Reviewer_PPJd · 2021-10-30

**Correctness:** 3
**Technical Novelty And Significance:** 3
**Empirical Novelty And Significance:** 3
**Recommendation:** 6
**Confidence:** 4

**Main Review:**

## Strength
This paper attempts to analyze one important problem of SVGD. Although this phenomenon has been identified in many previous works, this paper is the first one that theoretically quantifies the converged variance beyond the mean-field assumption. Qualitatively, the connection between MMD-descent and SVGD is intuitive (although I am still a bit confused about why introducing MMD-descent is necessary). Personally, I believe the theoretical analysis under the proportion limit is a significant contribution to the community, which is much closer to the actual empirical behaviour than the typical mean-field assumption. However, I still have several confusions.

## Weakness
My first question is why the connection between MMD-descent and SVGD is necessary? I understand that they share identical repulsive forces and similar driving forces, but they are still different. Thus, any finding from analyzing MMD-descent cannot be directly transferred to SVGD, no? For example, in section 4.1, proposition 2 revealed the large variance of the driving force in MMD-descent. However, you only analyze its effect on MMD-descent, not SVGD. Specifically, whether the driving force in SVGD has a large variance or its empirical effects are not analyzed. So what is the purpose of introducing MMD-descent and analyzing the large variance of the driving force?

Proposition 2 only confirms that the driving force in MMD-descent has a large variance. Is it still true for SVGD?

I also find that Section 4 and 5 are a bit disconnected. In Section 4, the argument is that the variance collapse problem is due to the deterministic bias of the driving force, which is confirmed empirically. However, the theoretical analysis in section 5 does not reveal why this bias causes the variance collapse problem. Also, I recommend that the author should add a sentence explaining where the deterministic bias assumption is used in the analysis.

If I understand correctly, the entire analysis is based on the Euclidean distance kernel assumption. However, there are many types of kernels. Do they still suffer from the variance collapse problem? In other words, do you think the variance collapse problem is due to the SVGD itself or the kernel associated with it?

For the empirical verification of (A2), how do you select $\epsilon_d$?


**Summary Of The Paper:**

This paper analyzed the curse-of-dimensionality problem of the vanilla SVGD with Euclidean distance kernel in a qualitative and quantitative way. Specifically, the author first built a connection of SVGD to MMD-descent, where they share identical repulsive forces with different driving forces (if Euclidean distance kernel is adopted). Then, the author argued that the variance collapse problem is rooted in (1) high variance and (2) the deterministic bias of the driving force, which were confirmed by sampling from the isotropic Gaussian. Quantitatively, the author analyzed the stationary variance of MMD-descent and SVGD with isotropic Gaussian under the proportional limit, which confirms the curse-of-dimensionality problem of SVGD.

**Summary Of The Review:**

Overall, this paper is written clearly and easy to follow. It also has significant theoretical contributions about the SVGD dynamics under the proportion limit. Although I am a little confused about the motivation of introducing MMD-descent, this paper has more merits than its drawbacks.

---

> ### Author Response · Authors · 2021-11-22
> **Response to reviewer PPJd**
>
> Thank you for the thoughtful review and detailed comments. We address the technical points below.
>
> **1. "why the connection between MMD-descent and SVGD is necessary?" "Proposition 2 only confirms that the driving force in MMD-descent has a large variance. Is it still true for SVGD?"**
>
> We introduce MMD-descent due to its structural similarity with SVGD, which allows us to formulate hypotheses on the cause of algorithmic bias that can be empirically verified for both algorithms. We agree with the reviewer that Proposition 2 does not directly apply to SVGD (as acknowledged in the footnote on page 4), and there is no rigorous justification that variance collapse in SVGD is entirely due to $S_1$. However, the intuition that $S_1$ may have higher fluctuation (and hence is more likely to be the problematic term) motivates us to design controlled experiments in Section 4.2, where we resampled either $S_1$ and $S_2$ in MMD-descent and SVGD.
> In particular, Figure 3(c) shows that even if the repulsive force $S_2$ is not resampled (i.e., still "biased"), SVGD can estimate the variance accurately; whereas if the driving force $S_1$ is not resampled, then estimating $S_2$ using different schemes does not ameliorate the issue. This provides empirical evidence that the driving force in SVGD plays a significant role in the variance collapse.
>
> **2. "Section 4 and 5 are a bit disconnected."**
>
> We agree that our results in Section 4 and 5 are mostly complementary to each other (i.e., Section 5 is not fully built upon Section 4); we have added a short discussion in the beginning of Section 5 to clarify this point.  Specifically, the goal of Section 4 is to provide a *qualitative* understanding of the algorithmic bias of SVGD; while it is argued that the absence of particle resampling is problematic in SVGD, Section 4 does not predict whether such bias leads to under- or over-estimation of the dimension-averaged variance.
> On the other hand, Section 5 provides a *quantitative* analysis of the variance collapse phenomenon in the proportional limit. The analysis makes use of the deterministic nature of the SVGD update, which allows us to directly work with the fixed point condition (e.g., Equation 12). In addition, our heuristic modification in Section 5.3 demonstrates that a simple modification of the driving force $S_1$ leads to improved variance estimation (for the Gaussian case).
>
> **3. "Do other types of kernel suffer from the variance collapse problem?"**
>
> Indeed we focus on the Euclidean distance kernel, which covers many standard kernels (Gaussian RBF, IMQ, etc.) for SVGD -- we have revised the Introduction to highlight this setting. We would appreciate if the reviewer could point us to some non-Euclidean distance kernels that are also commonly used in SVGD.
> Our current analysis does not rule out the possibility that a special (non-Euclidean distance) kernel can mitigate or completely avoid the variance collapse problem. Designing such a kernel would be an interesting research direction.
>
> **4. "For the empirical verification of (A2), how do you select $\epsilon_d$?"**
>
> Assumption (A2) requires $d^{-1/2}\cdot\epsilon_d \to 0$ as $n,d\to\infty$. Therefore we directly report this quantity in Figure 3 (and Appendix A.2): we observe that the quantity decays to $0$ as $n$ increases. This does not require choosing $\epsilon_d$ explicitly.
>
> We would be happy to clarify any further concerns/questions in the discussion period.

---

### Official Review · Reviewer_du4P · 2021-11-02

**Correctness:** 3
**Technical Novelty And Significance:** 3
**Empirical Novelty And Significance:** 3
**Recommendation:** 6
**Confidence:** 3

**Main Review:**

# Pros
- The paper is well written and easy to follow.
- There is a complete set of experiments to support the authors' argument.
- Theory in simplified settings is provided to justify the understanding.

# Cons
- The presented theory is a bit unsatisfactory in the sense that (1) the target data distribution is too simplified and (2) the proposional assumption ($d/n \to \gamma > 1$) weakens the theory for explaining the experimental results. In specific:
   *  Based on my understanding of the proof (note that I did not look in detail), (1) greatly simplifies the argument in that the studied problem is basically 1-dimensional. I believe the theory could be much more interesting if (1) can be relaxed to allow general PSD matrix as the covariance of the target distribution.
   * When I went through the paper, which is enjoyable process, I did not realize the necessity of  (2) until Section 5. In particular, the experimental observation, e.g., Figs 1 and 2, does not seem to satisfy (2), and in these experiments the variance collapse issue still exists in SVGD. It feels like (2) is invented only for the purpose of making up a theory.



**Summary Of The Paper:**

This work studies the variance collapse phenomenon of SVGD. By comparing to MMD-descend, the authors argue that the driving force of SVGD suffers from a bias caused by reusing data, and thus tends to underestimate the variance of the target distribution. Theory are developed in the setting of estimating standard Gaussian with a proposal limit (i.e., $d/n \to \gamma$), and explains the understanding in the overparameterized/high-dim setting (i.e., $\gamma > 1$). Experiments are also conducted to verify the understanding. Finally, motivated by the understanding, new algorithm is proposed to fix the issue of SVGD by damping the driving force term in SVGD.

**Summary Of The Review:**

First of all I am not an expert on SVGD thus I tend to be conservative in evaluating the paper.
Overall I think the paper is written very well & I have enjoyed much going through it. The problem studied in the paper, the variance collapse of SVGD, seems to be important and interesting in my perspective. I think the authors have made sharp observation towards understanding this issue & how to mitigate it. The main pitfall of this work, in my perspective, is the theory part; as explained above, I think the presented theory has quite some gaps from the experimental findings.

I am leaning to weak acceptance for now. Authors feedbacks are welcome as I may have missed some important points.

---

> ### Author Response · Authors · 2021-11-22
> **Response to reviewer du4P**
>
> Thank you for the helpful comments. We address the technical concerns below.
>
> **1. "the target data distribution is too simplified"**
>
> We acknowledge the scope of our theoretical analysis in the main text, and agree with the reviewer that extending our results to more general target distribution is an important direction. A first step towards a general result is provided in the appendix (Proposition 8), where we consider potentials that have Gaussian tail outside of a compact ball. On the empirical side, we also observe similar variance collapse phenomenon in other settings (BNN, MoG, etc.). We make the following comments.
> - As noted in the beginning of Section 5.2, our primary goal is to demonstrate a *negative result*: we do not claim that SVGD underestimates the variance in all settings; instead, we show that variance collapse is present even in learning the simplest target distribution.
> - Our particular choice of target distributions is not arbitrary for the following reasons.
> (i) One of the earliest observations of the variance collapse phenomenon considered the unit Gaussian target [Zhuo et al. 2017], yet there is no theoretical understanding of the underlying mechanism even in this simple case.
> (ii) isotropic Gaussian prior is widely used in Bayesian inference (if not the most common). Hence it is not unnatural to expect that target potential in some real-world problems exhibits quadratic growth outside of some radius.
> - Our current analysis in the proportional limit relies on Taylor expansion of the kernel matrix (analogous to existing works on nonlinear random matrix theory), but we do not reduce the studied problem to 1-dimension. In the case of general covariance, we speculate that following [El Karoui et al. 2010], we can Taylor-expand the kernel around the normalized trace $\text{tr}(\Sigma)$ and derive similar results; we leave the precise characterization for the general setting as future work.
>
> **2. "the proportional assumption weakens the theory for explaining the experimental results"**
>
> Indeed our theoretical results in Section 5 assume the proportional asymptotic limit. This setting is motivated by the observation that many modern Bayesian inference tasks are high-dimensional, and thus SVGD is often employed in the $d>n$ regime.
> The proportional scaling is a very common setting in the high-dimensional statistics literature, especially in the recent line of works on the precise asymptotics of overparameterized models (e.g., [1-3]). These precise results usually provide an accurate description under *moderate data size*, which is also true in our case (observe that our theoretical prediction matches the empirical finding for $d\asymp 100$). With some extra work, we believe that our asymptotic characterization can be translated to non-asymptotic (i.e., finite $n,d$) high probability guarantees.
>
> [1] Thrampoulidis et al, 2016. Precise error analysis of regularized M-estimators in high-dimensions.
> [2] Sur and Candes, 2018. A modern maximum-likelihood theory for high-dimensional logistic regression.
> [3] Mei and Montanari, 2019. The generalization error of random features regression: precise asymptotics and double descent curve.
>
> We would be happy to clarify any further concerns/questions in the discussion period.

---

### Official Review · Reviewer_r5Ry · 2021-11-04

**Correctness:** 4
**Technical Novelty And Significance:** 3
**Empirical Novelty And Significance:** 3
**Recommendation:** 6
**Confidence:** 3

**Main Review:**

I really enjoyed the paper and learned a lot from it. I was not familiar with this issue, but I appreciated the dedication and the in-depth study: it is rare to see a paper dedicated to "understanding" these days. All my congrats to the authors also for the nice writing and the nice figures.

My acceptance is only "weak" though since I would have loved to see the following:

1) a more thorough investigation of the "fixed SVGD": this only takes a page, and though some additional results are presented in the appendix I feel it deserves more space, or potentially also a few weeks of "rethinking" by the authors. Do you maybe have some ideas on how to make this section more complete? For instance, it would be great if the authors were able to show how damping provably fixes variance collapse in the high-dim Gaussian case. Examples are often underrated, yet these specified proofs often highlight nice ideas and concepts.

2) I feel that the method under investigation is only SVGD, yet many other methods exist to estimate probability densities. I think the paper would really profit from a comparison with other methods, just to set the right context.

All the rest is really good, though it maybe can be shortened a bit to leave space to more experimental results or more results (potentially a consistency theorem) for damped SVGD.

**Summary Of The Paper:**

This paper provides an understanding of the variance collapse phenomenon of SVGD. The paper first (1) introduces the reader to the most important concepts and phenomena, then (2) gives an explanation for why this problem occurs, thanks to a comparison with an accurate (yet computationally intensive) algorithm they call MMD-descent. Finally (3) the paper shows how to fix SVGD with damping. The paper provides experiments and theory, nicely combined.

**Summary Of The Review:**

A great paper, well written. The topic is nice. Some things can be improved (taken a step forward) to make the paper truly valuable also for practical uses.

---

> ### Author Response · Authors · 2021-11-22
> **Response to reviewer r5Ry**
>
> Thank you for the valuable comments, which helped us improve the manuscript. We address the technical points below.
>
> **1. A more thorough investigation of the "fixed SVGD".**
>
> Thank you for the suggestion. We have made the following updates to the manuscript.
> - In Appendix A.3 we included an additional Bayesian logistic regression (BLR) experiment for our modification in Section 5.3. We observe that the modified update generates more diverse samples, but it does not always accurately estimate the dimension-averaged variance -- in fact, the algorithm may slightly *overestimate the target variance* in the small particle regime (also observed in Figure 5 in the main text). This is not entirely surprising because the correction term is derived for the isotropic Gaussian target. We also reported the MMD between the SVGD particles and the (approximate) ground truth particles obtained from Langevin Monte Carlo.
> - In Appendix C.2 we included a short sketch of how the heuristic modification is derived. Roughly speaking, the damping term in the driving force $S_1$ is added such that the fixed point particles exactly match the ground-truth dimension-averaged variance under assumptions (A1-4).
>
> We remark that the goal of our work is to highlight the variance collapse of SVGD and precisely characterize the phenomenon in simple settings, and we do not claim to provide a general and realistic solution. In particular, our modified algorithm in Section 5.3 is derived *only for unit Gaussian target* (as commented in the footnote on page 9), which is the reason that not much focus is given in the main text.
> We hope that the new BLR experiment (Appendix A.3) helps clarify the limitation of this modification in more general settings.
>
> **2. Comparison with other methods to estimate probability densities.**
>
> This is a good point -- indeed our current analysis only covers two specific interacting particle algorithms (SVGD and MMD-descent), as commented in Section 2.2. Following your suggestion, we included a more thorough discussion on other sampling methods in Appendix A.4, with focus on Langevin Monte Carlo (LMC) and Herding.
> More specifically, we show that LMC does not suffer from variance collapse in high dimensions, due to the existing quantitative convergence guarantee. In particular, convergence rate of LMC under the $\chi^2$-divergence implies that after $\tilde{O}(d)$ iterations, *one particle* suffices to accurately estimate the dimension-averaged variance of the target distribution, under the same asymptotic setting in Section 5. This indicates that variance collapse is a not ubiquitous phenomenon in sampling from high-dimensional distributions; instead, it is likely that the issue is specific to interacting particle algorithms like SVGD.
>
> We would be happy to clarify any further concerns/questions in the discussion period.

---

### Official Review · Reviewer_SH8F · 2021-11-08

**Correctness:** 4
**Technical Novelty And Significance:** 3
**Empirical Novelty And Significance:** 3
**Recommendation:** 8
**Confidence:** 4

**Main Review:**

Strengths:
1. It is a good paper to analyze the limit and weaknesses of SVGD, especially the variance collapse issue on high-dimensional distributions. It is worth reading for the researchers working on Bayesian inference.
2. The proposed MMD-descent motivates some modifications to the original SVGD.
3. The experiments on toy datasets perfectly validate the claims of this paper.

Weaknesses:
1. Only toy datasets are used to validate the proposed approach. I would like to see a more practical experiment. E.g., in the original SVGD paper, Bayesian logistic regression and Bayesian Neural nets (BNN) on practical datasets are discussed. I would like to know on real scenario any advantage could be found with the proposed method.

**Summary Of The Paper:**

In this paper, the authors analyze the underestimation issue of stein variational gradient descent (SVGD), and propose the maximum mean discrepancy (MMD) descent. From the perspective of the decomposition of the gradient term (driving force and repulsive force), this paper suggested to use another driving force term instead of the original one in SVGD. But the new driving force makes MMD-descent impractical since it depends on an intractable integral of the desired distribution $p$. In addition, the paper identify the log derivative driving force as the problematic term in SVGD, and propose a modified SVGD with particle resampling. They also argue that the proportional asymptotic limit is more relevant to understanding the variance collapse phenomenon. The theoretical dimensional analysis of SVGD on Gaussian also suggested another modified (damped) SVGD.

**Summary Of The Review:**

Overall, this is a good paper with solid analysis. It should be of interest to the area of Bayesian inference. But I think more practical validation should be addressed.

---

> ### Author Response · Authors · 2021-11-22
> **Response to reviewer SH8F**
>
> Thank you for the positive evaluation and thoughtful feedback. We address the technical comments below.
>
> **1. "more practical experiment ... with the propose method".**
>
> Thank you for the suggestion. We included an additional experiment on Bayesian logistic regression (BLR) in Appendix A.3. The new experiment (together with the BNN results in the same section) illustrates that the proposed modification generates more diverse samples; that said, we also note that this method is not guaranteed to accurately estimate the dimension-averaged variance beyond the Gaussian setting.
>
> We emphasize that the focus of our work is to highlight the variance collapse of SVGD and precisely characterize the phenomenon in simple settings, and we do not claim to provide a general and realistic solution beyond the Gaussian case. More specifically,
> - The runtime of the particle resampling procedure scales quadratically with the number of iterations (as commented in Section 4.2), and thus it is *computationally intractable* in many practical Bayesian inference problems, where the algorithm might take a large number of steps to converge.
> - The heuristic modification in Section 5.3 is derived for the unit Gaussian target in the proportional limit, as a *proof-of-concept* that damping the driving force can alleviate the variance collapse problem. However, as commented in the footnote on page 9, it is not the case that the modified update always accurately estimates the dimension-averaged variance in more general settings.
> In fact, in the new BLR experiment in Appendix A.3, we see that the (Gaussian) correction term might slightly overestimate the target variance in the small particle regime (also observed in Figure 5 in the main text). We will further emphasize on this limitation in the manuscript.
> - Since our goal is to understand the variance collapse phenomenon, we do not compare the test loss/accuracy as in the original SVGD paper. It is possible that accurate estimation of the posterior variance does not lead to optimal test performance (which is orthogonal to our investigation).
>
> We would be happy to clarify any further concerns/questions in the discussion period.

---

### Author Response · Authors · 2021-11-22
**General reponse**

We thank the reviewers for their time and valuable feedback, which helped us improve the paper significantly. We are glad to see that all the reviewers have positive opinions regarding our work and find it interesting. We reply to the reviewers’ feedback as separate comments below, and we have also posted a revised manuscript. If there are further questions/comments/suggestions, we would be happy to address them in the discussion period.

---

### Decision · Program_Chairs · 2022-01-20

**Decision:**

Accept (Poster)

**Comment:**

This paper proposes new analysis on Variance Collapse of SVGD in High Dimensions. The analysis provides some new insights despite of some limitations.